

# Classification of mechanisms, Climatic Context, Areal Scaling, and Synchronization of floods: the hydroclimatology of floods in the Upper Paraná River Basin, Brazil

Carlos Lima[1], Amir AghaKouchak[2], and Upmanu Lall[3]

[1]Civil and Environmental Engineering, University of Brasilia, Brasilia, Distrito Federal, Brazil.
[2]Civil and Environmental Engineering, University of California, Irvine, Irvine, California United States.
[3]Earth and Environmental Engineering, Columbia University, New York, New York, United States.

*Correspondence to:* Carlos Lima (chrlima@unb.br)

**Abstract.** Flood is the main natural disaster in Brazil, causing substantial economic damage and losses of lives. Recent studies suggest that some extreme floods in different parts of the world do not appear as random as they are represented in traditional flood frequency analysis (FFA), but result from a causal chain, where exceptional rain and floods in basins from different sizes are related with large scale anomalies and persistent patterns in the atmospheric and oceanic circulations. Moreover,
floods result from different generating mechanisms or are subject to temporal changes in the forcing mechanisms and surface conditions, which violates the common homogeneity and stationary assumptions in FFA. An Eulerian-Lagrangian model of ocean-atmosphere circulation would ideally be needed to test a causal chain hypothesis. However, some progress may be possible through empirical data analysis. Here we seek to advance the traditional statistical flood analysis, through understanding the flood generating mechanisms including large scale patterns of the ocean and atmospheric circulation. We outline a
methodological framework based on the Self-Organizing Map (SOM) clustering that allows linking large scale processes to local scale observations. The proposed methodology is applied to flood data from several sites in the flood prone Upper Parana River Basin (UPRB) in southern Brazil. The SOM clustering approach is employed to classify the six-day rainfall field over UPRB into four categories, which are then used to classify floods into four types based on the spatio-temporal dynamics of the rainfall field prior to the observed flood events. An analysis of the vertically integrated moisture fluxes, vorticity and high level
atmospheric circulation revealed that these four clusters are related to tropical and extra-tropical processes, including the South America low-level jet (SALLJ), extra-tropical cyclones and the South Atlantic Convergence Zone (SACZ). Persistent anomalies in the sea surface temperature fields in the Pacific and Atlantic oceans are also found to be associated with these processes. Floods associated with each cluster present different patterns in terms of frequency, magnitude, spatial variability, scaling and synchronization of events across the sites and subbasins. These findings and the methodological framework proposed in this
study provide new insights for understanding causes of floods around the world and are a step forward to improve flood risk management, interpreting statistical assessments and short-term flood forecasting.



## 1 Introduction

The underlying assumptions of homogeneity, stationarity and randomness assumed in traditional flood frequency studies, have been questioned in a substantial number of studies (e.g. Jain and Lall (2001); Smith et al. (2011); Hirschboeck et al. (2000); Milly et al. (2002); Alila and Mtiraoui (2002); Kwon et al. (2008); Lima et al. (2015); Merz et al. (2014); Neiman et al. (2011); Seo et al. (2012); Villarini et al. (2009, 2013); Vogel et al. (2011); Westra and Sisson (2011)). To make progress on understanding and modeling the real world flood process one needs to better understand how the complex forms of interaction among weather, climate, hydrology and basin attributes and antecedent conditions evolve over time and space to produce a given flood.

Historically, flood studies have followed two distinct research lines: hydrometeorology of floods and flood frequency analysis. Flood hydrometeorology focuses on understanding: i) hydrodynamics of the rainfall-runoff process during flood events; ii) spatial structure of local rainfall events that are associated with floods; iii) soil-atmosphere response and large scale circulation patterns associated with the forecast and diagnostic of rainfall events. Some examples can be seen in Maddox (1983); Kunkel et al. (1994); Pal and Eltahir (2002); Schumacher and Johnson (2005); Amengual et al. (2007); Viglione et al. (2010); Li et al. (2013). There is also an extensive literature related to the statistical analysis and modeling of flood frequency from local and regional data of rainfall, streamflow and water basin attributes, including non-stationary approaches (e.g. Thomas and Benson (1970); Stedinger et al. (1993); Stedinger and Cohn (1986); Kwon et al. (2008); Kroll and Stedinger (1998); Lima and Lall (2010)).

In this study, we investigate floods in the Upper Paraná River Basin (hereafter, UPRB) in southern Brazil focusing on the hydroclimatology framework and understanding the flood generating mechanisms (Hirschboeck (1988)). The overarching goal is to link frequency of flood events to flood generating mechanisms to provide a better understanding of the underlying processes. The stationary assumption in most flood frequency studies is enriched by a formal consideration of the physical mechanisms responsible for generation of extreme floods. This includes a recognition of the natural climate variability associated with persistence and oscillatory regimes (e.g., El Niño) across different time scales (e.g., inteannual, decadal, etc) as well as climatic changes in response to anthropogenic changes in atmosphere, soil and land use.

Many studies have investigated the interactions between basin attributes and atmospheric circulation leading to extreme or exceptional floods. However, there is a gap of knowledge on how the evolution of large scale climate modes at the interannual scale changed the *chances* of local precipitation and soil humidity and consequently the probability distribution of floods. It is argued that the frequency of flood events is very sensitive to modest changes in climate (Knox (1993)). We explore the Hirschboeck's hypothesis (Hirschboeck (1988)) that *exceptional floods in basis of all sizes* could be related to anomalies in the large scale atmospheric circulation. This flood hydroclimatology perspective has been applied to identify the moisture transport and large scale climate patterns associated with floods in the United States (Hirschboeck (1988); Budikova et al. (2010); Nakamura et al. (2013); Lu and Lall (2016); Mallakpour and Villarini (2016)), Europe (Prudhomme and Genevier (2010); Jacobeit et al. (2003); Bárdossy and Filiz (2005); Lu et al. (2013)) and other parts of the world (Kahana et al. (2002)). However, such flood studies focused in South America are nonexistent to our knowledge.



Intuitively, a rainfall system that persists over a given locale with a continuous and sufficient supply of moisture (from advection and recycling) has a high likelihood of generating an exceptional flood, since at some point this system will saturate soil moisture. For sufficiently large drainage areas an extreme flood may require an external flux of advective moisture, i.e., local convective processes may not tend to produce exceptional floods in these basins. Moreover, such an influx of large scale

advective moisture maintain an increased potential for large floods as the drainage area and return period increase. Hirschboeck (Hirschboeck (1988); Hirschboeck et al. (2000)) notes that the scale of convective storms that can generate intense short rainfall is typically of some km$^2$ and is therefore it is unlikely that such convective processes are the main source of exceptional floods over large areas. On the other hand, mesoscale convective systems (MCS), such as convective complex (MCC) and squall lines, tend to cover large areas and persist for several hours and are sources of heavy rainfall in some regions of the USA

(Schumacher and Johnson (2005, 2006)) and also Brazil (Salio et al. (2007); Zipser et al. (2006); Durkee and Mote (2009); Durkee et al. (2009); Marengo et al. (2012)), in particular the MCCs to the east of the Andes that impact the La Plata Basin. However, there is evidence (Maddox (1983); Corfidi et al. (1996)) that the maintenance and development of such systems is related to large-scale atmospheric circulation features. Thus, tropical and extratropical cyclones and associated fronts become important in the production of extreme rainfall over large areas and are directly related to atmospheric circulation patterns of

large scale and with storm paths or well defined regions of moisture transport in the atmosphere.

We explore extreme floods in UPRB through a hydroclimatic analysis of flood series across 33 nested-basin sites with drainage areas ranging from 2,588 to 823,555 km$^2$. The spatio-temporal dynamics of daily rainfall over the basin in the days that preceded the largest flood events is analyzed and classified into clusters of similar patterns based on a Self-Organizing Map (Kohonen (2001)) clustering algorithm. This way, we intend to take into account the persistence and alignment of the

storm path with the drainage basin that produces a given flood. The associated large scale atmospheric circulation for each cluster is then analyzed in terms of moisture transport and convergence, high level circulation and vorticity. Teleconnections with the Atlantic and Pacific oceans are evaluated by composite analysis of the sea surface temperature (SST) field. For each rainfall cluster, the attributes (frequency, magnitude, scaling and synchronization) of floods across UPRB are analyzed in order to produce and characterized a typology for floods in the region according to the dynamics of rainfall patterns and associated

atmospheric circulation. The paper is organized as follows. In the next section we present the region of study and data. In section 3 we introduce the clustering algorithm. In section 4 we present the results and finally in section 5 we offer a summary and discussion.

## 2 Region of Study and Hydroclimate Dataset

### 2.1 The Upper Paraná River Basin, streamflow and rainfall dataset

The Upper Paraná River Basin is located in southern Brazil (Fig. 1) and is part of the La Plata basin, which is the second largest basin in South America after the Amazon basin. It concentrates a large population of Brazil and is of utmost importance for the country in terms of flood control, hydropower generation and agriculture. The rainfall season over UPRB is mostly marked by a peak during the austral summer (summer monsoon system) related to the South American monsoon system (SAMS) and





associated South Atlantic Convergence Zone (SACZ, see Barros et al. (2000); Jones and Carvalho (2002); Berbery and Barros (2002); Carvalho et al. (2004); Marengo et al. (2012)), particularly in the region north of 20°S, where the monsoon system is the dominant forcing (Berbery and Barros (2002)). Rainfall interannual variability has been associated with SST anomalies in the Tropical Pacific and South Atlantic oceans (Grimm et al. (1998); Robertson and Mechoso (2000); Doyle and Barros

(2002); Grimm (2003, 2004); Grimm et al. (2000); Cardoso and Dias (2006); Chaves and Nobre (2004); Jorgetti et al. (2014)). Intra-seasonal and decadal variability of rainfall and streamflow have been also the subject of many studies (Carvalho et al. (2004); Robertson and Mechoso (2000); Paegle and Mo (2002); Robertson et al. (2001); Zhou and Lau (2001)). Most of the moisture that reaches UPRB is from the Amazon region (Drumond et al. (2008); Carvalho et al. (2011)), and the rainfall mechanisms are also associated with Mesoscale Convective Systems (MCSs) along the South-American low-level jet (SALLJ)

(Velasco and Fritsch (1987); Marengo et al. (2004); Salio et al. (2007)) and transient systems related to extratropical cyclones and cold fronts (Mendes et al. (2007); Silva and Ambrizzi (2010)). El Niño events have also been linked to extreme rainfalls and floods in UPRB (Camilloni and Barros (2003); Grimm and Tedeschi (2009); Muza et al. (2009); Cavalcanti et al. (2015); Antico et al. (2016)).

We use naturalized, mean daily streamflow data from 33 sites in UPRB (Fig. 1). These sites are located in strategic points to

provide the inflow into the main hydropower reservoirs in UPRB, which are used not only for generation of electrical energy but also for flood control, water supply and agriculture. The dataset is offered by the National Operator of the System (ONS), which defines the operational rules of all interconnected hydropower reservoirs in the country. The streamflow data is available from January/1931 to December/2013, but in order to be consistent with the availability period of the rainfall dataset, we perform all analysis considering the streamflow data restricted to the 1980–2013 period. The catchment basin areas range from

2,588 to 823,555 km$^2$.

We limit our analysis to the warm season (November through March), when most floods occur (Lima and Lall (2011)). For each site, we obtain partial duration series of floods by taking the values in which the daily flow exceeds a given threshold. In order to keep a relatively large number of flood events in each rainfall cluster, we set this threshold as the 70th empirical flood quantile for the warm season. We analyze only independent floods by declustering the series (Lang et al. (1999)) and taking

events with inter-arrival times larger than 15 days, which we believe is a consistent interval to guarantee independence between flood events considering the different rainfall mechanisms that cause floods in UPRB. After doing this procedure, we obtain dates and magnitudes of about 98 flood events for each of the sites in UPRB analyzed here.

Daily gridded rainfall data for the period 1980–2013 are provided by Xavier et al. (2016). These data consist of interpolated daily rainfall observations from 3625 rainfall gauges and 735 weather stations across Brazil available from different institutions

(INMET, ANA and DAEE). The interpolation schemes and validation procedures are described in Xavier et al. (2016). The rainfall data is delimited by the UPRB boundary as shown in Fig. 1. For each grid point, daily anomalies of rainfall are obtained after removing, from the observed value, the long term monthly mean (with respect to the day being evaluated) for that grid point based on the 1980–2013 period.





## 2.2 Moisture Fluxes, Vorticity, Upper Level Winds and Sea Surface Temperature

Mean daily data of vertically integrated moisture fluxes and the associated divergence field (Evaporation - Precipitation along an atmospheric column), low and high level relative vorticity and high level (500 mb) winds are obtained from the ERA-Interim reanalysis data (Dee et al. (2011)). It covers the period from 1980 to 2013 and are retrieved for the region defined by

$15°$N-$60°$S and $270°$W-$330°$W.

We also use daily SST data from the ERA Interim global sea surface temperature archive for the 1980–2013 period. Daily SST anomalies for each grid point are calculated by subtracting, from the observed value, the monthly mean for that grid point and related month based on the 1980–2013 period. The SST field is delimited by the region $30°$N-$80°$S and $210°$W-$20°$E.

All dataset are interpolated for a grid of $2.5°$ x $2.5°$.

## 10  3   Technical approach for Rainfall Clustering

A flood event, defined as an elevation of river stage above its bank height, can have a duration of a few minutes and a spatial extend of a few square kilometers, while other flood events can last for months with spatial scales that can exceed $10^6 \, \mathrm{km}^2$. A large number of flood studies have focused on the understanding of physical processes associated with floods in basins of small scale due to the ease of observing critical events in these basins, while over large areas the focus has been on the problem to

predict flood quantiles, with less emphasis on the understanding of the physical mechanisms associated with extreme floods. For instance, the relation of soil moisture and a given rainfall event in producing some floods over small areas and homogeneous soils is relatively easy to evaluate. On the other hand, the problem becomes considerably more complicated as we consider large basins, with drainage areas over $10^4 \mathrm{km}^2$, since i) the potential of a high heterogeneity in the initial soil moisture field is high and ii) the location and direction of the storm path along the basin leads to a significant heterogeneity in the spatial and

temporal distribution of the rainfall event. Since the influx of large scale advective moisture may be a particular factor to overlie the initial heterogeneities on the surface conditions for larger basins, we will implicitly assume here that the spatio-temporal variability (basically magnitude, persistence and alignment of the storm path with the drainage basin) of rainfall is the key factor of producing floods across the UPRB sites evaluated in this work.

Let us assume then that the information regarding the spatio-temporal patterns of rainfall associated with the major flood

events is contained in a rainfall dataset represented by a matrix $\mathbf{X} = [\mathbf{x}_1 \, \mathbf{x}_2 \ldots \mathbf{x}_T]$, where $\mathbf{x}_t$ is a column vector containing all the relevant information about the spatial variability and persistence of daily rainfall over UPRB along days $t-\tau$, $t-\tau-1$, $\ldots$, $t$, for some time delay $\tau$. $T$ is the total number of effective days during the austral warm season (November–March) over the 1980–2013 period. Our goal then is to extract information about $\mathbf{X}$ through clustering. We will adopt the Self-Organizing Map (SOM) approach to cluster rainfall information as expressed in $\mathbf{X}$. SOMs are a particular case of competitive neural networks

and have been developed by the machine learning community in the 1990's (Kohonen (2001)) for cluster analysis and classification. They have been successfully applied to find clusters in climate systems (e.g. Cavazos (2000); Hewitson and Crane (2002); Johnson et al. (2008); Lee and Feldstein (2013); Bao and Wallace (2015); Li et al. (2015); Mioduszewski et al. (2016); Xu et al. (2016); Li et al. (2016)). An extended review of applications in Climate Science is provided by Liu and Weisberg





(2011). SOMs are also known as Kohonen neural networks and the basic idea is to obtain a 2-d topology consisting of nodes (or neurons) that are associated with the input space $\mathbf{X}$, preserving yet its topological features.

For the sake of clarity and understanding of the SOM properties and tuned parameters used in this work, we describe here the key aspects of SOM. We refer the reader to Kohonen (2001) for more details about SOM. Let us assume that we have

$K$ neurons, then initially $K$ representatives (or prototypes, synaptic weight vectors, reference vectors) are randomly chosen from the input space $\mathbf{X}$ and associated with the $K$ neurons. An input vector $\mathbf{x}_t$ is randomly selected from the data set $\mathbf{X}$ and the Euclidean distance between $\mathbf{x}_t$ and each representative $\mathbf{m}_k$, $k = 1, \ldots, K$, is computed. The neuron whose representative yields the smallest distance to $\mathbf{x}_t$ is the winner neuron $k^*$ or Best-Matching Unit (BMU):

$$k^* = \arg \min_k \{||\mathbf{x}_t - \mathbf{m}_k||\}. \tag{1}$$

In the next step, the neurons that are neighbors (neighborhood set) of the winning node $k^*$ are found based on the Euclidean distance and a given threshold $c$. The representatives corresponding to each grid neighbor of the wining neuron $k^*$ are then updated according to the rule:

$$\mathbf{m}_k \leftarrow \mathbf{m}_k + \alpha \cdot (\mathbf{x}_t - \mathbf{m}_k), \ k \in \mathcal{N}_c(k^*), \tag{2}$$

where $\alpha$, $0 \leq \alpha \leq 1$, is the so-called learning rate and $\mathcal{N}_c(k^*)$ denotes the set of points in the neighborhood of $k^*$ given the

parameter $c$. The process is then arbitrarily repeated a large number of times (*epochs*), since there is no explicit error criterion to minimize (Lee and Verleysen (2007)).

Variants of the update rule in equation (2) include a time varying learning rate $\alpha$ and weighted distances based on the proximity of $\mathbf{m}_k$ and the winning neuron $\mathbf{m}_{k^*}$:

$$\mathbf{m}_k \leftarrow \mathbf{m}_k + \alpha(j) \cdot h(||\mathbf{m}_k - \mathbf{m}_{k^*}||) \cdot (\mathbf{x}_t - \mathbf{m}_k), \ k \in \mathcal{N}_c(k^*), \tag{3}$$

where $\alpha(j)$ is the learning rate at epoch $j$ and $h(||\cdot||)$ is a neighborhood function around the winner neuron $k^*$. Common functions for $\alpha(j)$ include the linear, power and inverse functions with a decrease rate over time. A common function for $h(||\cdot||)$ is the Gaussian kernel:

$$h(||\mathbf{m}_k - \mathbf{m}_{k^*}||) = \exp\left\{-\frac{||\mathbf{m}_k - \mathbf{m}_{k^*}||}{2 \cdot \sigma^2}\right\} I_{k \in \mathcal{N}_c(k^*)},$$

where $\sigma$ is the width of the kernel (or neighborhood radius) and $I$ the indicator function.





In the batch version of the SOM, instead of presenting each time a single data vector, the entire data set $\mathbf{X}$ is presented to the SOM before any weights are updated and the BMU $\mathbf{m}_{k^*}$ is obtained for each input data $\mathbf{x}_t$ at each epoch, so that each data vector $\mathbf{x}_t$ will belong to a given neuron and the new neurons are updated as:

$$\mathbf{m}_k \leftarrow \frac{\sum_{t \in \mathcal{N}_c(k)} w_t \mathbf{x}_t}{\sum_{t \in \mathcal{N}_c(k)} w_t}, \tag{4}$$

where the weight function $w_t$ can be a rectangular function, which is equal to 1 for the neighbors of $\mathbf{m}_k$ and 0 otherwise, or be a smooth function $h(||\mathbf{m}_t - \mathbf{m}_k||)$. In this sense, each new neuron is a weighted average of the data samples that belong to its neighborhood neurons.

For a given number of neurons $K$, learning rate $\alpha$, threshold $c$ and fixed number of epochs, the trained SOM can encode any point $\mathbf{x}_t$ by giving the index $k$ of the closest neuron $\mathbf{m}_k$, where the distance is computed similarly to equation (1). In this way,

each data point of the entire dataset of rainfall information $\mathbf{X}$ can be assigned (or clustered) into one of the categories $1, \ldots, K$.

The final embedding of $\mathbf{X}$ can be evaluated by the mean quantization error (MMQE) of the SOM, which essentially measures the average distance of each input $\mathbf{x}_t$ to its representative in the output space:

$$\text{MMQE} = \frac{1}{T} \sum_{t=1}^{T} ||\mathbf{x}_t - \mathbf{m}_{\mathbf{x}_t}||, \tag{5}$$

where $\mathbf{m}_{\mathbf{x}_t}$ refers to the best matching unit of the corresponding $\mathbf{x}_t$.

In order to capture the spatio-temporal dynamics of the rainfall field over UPRB, including the information of antecedent rainfall for a given day $t$ of the record, we will concatenate the rainfall field over a time window $\tau = 5$ days:

$$\mathbf{x}_t = \left[\mathbf{r}_{t-5}\ \mathbf{r}_{t-4}\ \mathbf{r}_{t-3}\ \mathbf{r}_{t-2}\ \mathbf{r}_{t-1}\ \mathbf{r}_t\right]' \tag{6}$$

where $\mathbf{r}_t$ is a row vector representing the observed rainfall field over the Upper Paraná River Basin (Fig. 1) at day $t$, with dimension 1178 (number of grid points), so that $\mathbf{x}_t$ has dimension 7068.

It is interesting to note that as $\tau$ increases, the number of dimensions of $\mathbf{x}_t$ increases as well and the associated rainfall pattern may not be necessarily connected with the flood events. Based on the results discussed in the next section and the lifetime of about 3 days of extratropical cyclones (Simmonds and Keay (2000)) and 3 days of SACZ events (Carvalho et al. (2004)), we believe $\tau = 5$ days is an appropriate choice to extract the relevant information regarding the rainfall field during flood events.

We focus on the November–March daily rainfall, which is the main rainy season for UPRB. The rainfall dataset covers the period from January 1st 1980 to December 31st 2013 with a total of 5143 data points. After concatenating the rainfall field as explained in equation (6), the number of data points reduces to 5138, starting now in January 6th 1980 and ending in December 31st 2013. This results in a 5138 x 7068 input data matrix $\mathbf{X}$ to the SOM.



## 4 Results

### 4.1 Rainfall Clustering

We choose a 2x2 hexagonal grid to define the SOM, which means that the rainfall field will be classified into $K = 4$ clusters. This choice is made primarily to associate a relatively large number of flood events in each rainfall cluster. The neighborhood radius $c$ is initially set as 3 and monotonically decreases to 1 (equivalent to 6 neighbors for a central neuron in an hexagonal grid) when the number of epochs is equal to 100. This is the so-called ordering phase, where a global order is achieved for the map (Kohonen (2001)). From 100 epochs $c$ is set to 1 (tuning phase). Since the SOM grid consists of four neurons, then only two neighbors will have the size of its neighborhood affected by $c$ (see Fig. 2 and related discussion). The weight function $h$ in equation (4) is the rectangular function. The total number of epochs is set to 1000, but we do not observe any significant difference in the mean quantization error (MMQE) after the first 200 epochs. At 1000 epochs we obtained MMQE = 777.69. We also evaluate MMQE for a 2x3 and a 3x3 hexagonal grids and observe that the values tend to oscillate around MMQE = 777.69 as a function of the number of epochs, so that any significant differences for the 2x2 grid are observed. The SOM clustering algorithm is obtained using a commercial Neural Network Toolbox (MATLAB (2014)).

Figure 2 shows the final SOM after 1000 epochs in terms of hits in each neuron (left panel) and neighbors and weight distances (right panel). The number of hits is almost evenly distributed among neurons 1, 2 and 3. Neuron 4 has almost the double of hits of the other neurons. Due to the hexagonal grid layout, neurons 3 and 4 are connected to all the remaining neurons, while neurons 1 and 4 are connected only to neurons 3 and 2 (right panel of Fig. 2). The shortest distance is between neurons 3 and 4 while the largest distance is between neurons 3 and 2.

The above analysis is complemented by looking at the weights of each neuron (Fig. 3), which basically contain the information about the rainfall anomaly field over UPRB from day $t-5$ to day $t$. Neuron 1 has a north-south seesaw pattern at day $t-5$ and progressively moves towards an homogeneous field, with a strong rainfall peak at day $t-3$ centered in the northeastern part of the basin. The north-south dipole structure returns stronger at day $t-2$ and persists until day $t$, but now with a decrease in the rainfall peak. At this point it is worth mentioning that the negative anomalies in the rainfall field do not necessarily imply absence of rainfall, but just that the rainfall in that specific grid point is below its long term monthly average. Neuron 2 starts with a somewhat homogeneous rainfall field at times $t-5$ and $t-4$, from which negative rainfall anomalies start in the southern part and cover approximately the entire basin at time $t$. Neuron 3 starts at time $t-5$ with a northeast-southwest dipole structure with positive anomalies in the southwest, which progress over time until almost the entire basin is covered by positive anomalies at time $t$. Neuron 4 has an homogeneous rainfall pattern over the entire basin, with negative anomalies from time $t-5$ to time $t$.

Combining the information from Figures 2 and 3, we observe that the rainfall field represented by neuron 4 likely reflects conditions close to the average rainfall pattern during the rainy season. It is somehow connected to the rainfall patterns expressed by neurons 2 and 3, which are also connected to neuron 1. The shortest distance is obtained between neurons 3 and 4, followed by the distances of neurons 1 and 3 and neurons 2 and 4.





Considering that each neuron represents a given state of the rainfall field during the course of 6 days, we estimate transition probabilities across the states and show them in Table 1. We note that there is a general tendency of the rainfall field to remain in its state (neuron), but the transition probabilities are different among neurons. Neuron 1 is more likely to transition to neuron 2, which is more likely to transition to neuron 4. Neuron 3 has the highest probability to transition to neuron 1, while neuron 4

will more likely stay at its own state, with just a small probability to transition to neuron 3. We discuss further and contextualize these transitions in the next section when we analyze the atmospheric circulation associated with each neuron.

## 4.2 Atmospheric Circulation, Moisture Transport and Sea Surface Temperature

The analysis of key atmospheric and ocean variables in each neuron class is conducted here through a composite analysis considering the days correspondent to each neuron class. In this sense, they will reflect the average conditions for day $t$ as

showed in Fig. 3. Given the persistence of these variables and the episodes of SACZ and SALLJ, we do not expect substantial changes in the patterns found for days $t-1$ through $t-5$.

The vertically integrated moisture flux and the associated divergence field (Evaporation - Precipitation along an atmospheric column) averaged over each neuron class is shown in Fig. 4. We can see this as a climatology of the moisture transport associated with the rainfall patterns indicated in Fig. 3. Neuron 1 shows an intense moisture transport from the Amazon region,

possibly associated with SALLJ episodes (Marengo et al. (2004)). The divergence field is negative in the northern portion of UPRB, suggesting intense rainfall along this region, and positive in the southern part (dry conditions), extending to 50°S. This dipole structure has been reported in several studies (e.g. Nogués-Paegle and Mo (1997); Díaz and Aceituno (2003); Liebmann et al. (2004)) and is also observed in the rainfall field associated with neuron 1 at time $t$ (bottom panel of Fig. 3). The circulation is similar to the pattern described by Nogués-Paegle and Mo (1997) for negative events, where convection in

the SACZ in enhanced and more likely to occur during El Niño episodes. The SALLJ is weak, consistent with other studies (Liebmann et al. (2004)), including model-based ones (Silva and Berbery (2006)).

The moisture transport in neuron 2 is dominated by a north-south meridional flow crossing the entire basin, with a relatively homogeneous convergence of moisture over the basin, resembling also the rainfall pattern at time $t$ for neuron 2 (bottom panel of Fig. 3). This pattern seems to be associated with a weaker SACZ and stronger SALLJ, as described in Nogués-Paegle and Mo

(1997) for positive events.

The moisture transport in neuron 3 is also affected by a strong SACZ and moisture fluxes from the Amazon region but, when compared with neuron 1 (Fig. 4), the divergence (or inhibited precipitation) is far south of the basin and covers a smaller area. The moisture divergence pattern is again similar to the rainfall field at time $t$ for neuron 3 (bottom panel of Fig. 3). Neuron 4 has a moisture transport pattern somewhat similar to that of neuron 2, but the origin of the fluxes are more associated

with the South Atlantic, with meridional fluxes west of the basin, and a less intense but still relatively homogeneous moisture convergence. This reflects the rainfall field for neuron 4 (Fig. 3) and is likely associated with the average conditions of moisture transport into the region (Doyle and Barros (2002); Carvalho et al. (2004)).

The dynamics of the moisture transport associated with each neuron class is complemented by analyzing the low level (850 mb) relative vorticity (Fig. 5), which can indicate zones of low pressure and cyclonic rotation. A distinguished pattern is found



for neuron 1, with negative relative vorticity or cyclonic rotation over the entire basin and positive relative vorticity centered around 60°W 30°S, which suggests dynamical forcing and upper level wave activity associated with neuron 1. This pattern has been identified in other studies (Liebmann et al. (1999); Robertson and Mechoso (2000)). Neuron 3 also shows cyclonic rotation (negative relative vorticity) in the southern part of UPRB, extending up to 30°S. Neurons 2 and 4 do not show any sign

of intense cyclonic flow over the basin.

     The high level (500 mb) atmospheric circulation and relative vorticity associated with each neuron class is shown in Fig. 6. Neuron 1 shows a strong trough in the upper level circulation that extends to the entire UPRB, with negative vorticity over the entire basin and positive vorticity southwestern of it. This pattern confirms our hypothesis that this neuron is also associated with upper level wave activities. Neuron 3 shows also a trough over the basin, but it is weaker and negative vorticity appears

only in the south. Neurons 2 and 4 show more a zonal kind of circulation south of 20°S, which resembles the climatology of high level circulation.

     Anomalies in the near surface air temperature associated with each neuron is shown in Fig. 7. Neurons 1 and 4 have, respectively, negative and positive anomalies that cover the entire UPRB. Neuron 3 has a sharp contrast of negative anomalies in the south and positive anomalies in the north, suggesting frontal activities. Neuron 2 has also a sharp contrast of anomalies

but with opposite sign as compared with neuron 3 and the pattern suggests that it results from the advection of moist and warm air from the Amazon.

     Potential SST persistent patterns associated with each neuron are analyzed here by passing a 15-day high frequency filter on the daily SST anomalies, which are calculated by subtracting, from the daily SST, the average of the correspondent month for the January/1980 – December/2013 period. The results are shown in Fig. 8. Neuron 1 and neuron 3 show both positive

anomalies in the El Niño region, in the central Pacific and Tropical Atlantic. A dipole kind of structure appears in both neurons along the southern coast of South America but they are out-of-phase. The negative SST anomalies off the South America coast associated with neuron 1 have been identified in other studies (Doyle and Barros (2002)) during SACZ activities and it is not clear whether they are a response to the reduced income radiation from the intense rainfall band that extends from the Amazon to the South Atlantic or they are in fact acting to produce the observed circulation pattern. The SST pattern of neuron 3 is

similar to that of neuron 1, except that the anomalies off the South America coast near 30°S are positive, which is somehow consistent with the positive rainfall anomalies in the southwestern part of the basin (3) and the results of Doyle and Barros (2002). Neurons 2 and 4 show a similar pattern along the South Atlantic, with positive anomalies in the tropics, negative in the subtropics and positive south of around 40°S. The SST pattern in the Pacific ocean for neuron 2 is diffuse, with no remarkable feature. Neuron 4 shows positive and negative anomalies that intercalate across the Pacific, with negative anomalies along the

El Niño region. The SST anomalies in the Atlantic for neuron 2 are very similar to those observed for neuron 1.

     Combining all this previous analysis, we can shed some light on the transition probabilities, hits and connectivity among neurons as displayed in Fig. 2 and Table 1. Neuron 4 has individually the most hits and likely reflect the average circulation during the wet season, with a strong persistence but reduced SACZ activities. Eventually it precedes neuron 3 (probability = 11%) and most likely succeeds neuron 2 (probability = 22%), which is somehow expected given the rainfall pattern as shown





in Fig. 3 and the atmospheric circulation and SST anomalies in Figures 4 to 8. Neuron 2 has also a slightly probability (11%) of precede neuron 3 and most likely (probability = 35%) succeeds neuron 1.

When we connect these results with the transition probabilities in Table 1, we can describe the most probable sequence of rainfall states. The dynamical forcing and active SACZ of neuron 1 is most likely preceded by neuron 3 (probability = 18%),
which is marked by active SACZ, high level waves and cold fronts, and will most likely be followed by the rainfall pattern of neuron 2 (probability = 35%), which is somewhat coherent with the surface air temperature march as inferred from Fig. 7. Neuron 4 will most likely be followed by neuron 4 (Fig. 3). Neurons 1 and 4 are not connected and the transition probabilities between them are practically zero. In summary, the most likely sequence of neuron transitions is: $3 \rightarrow 2 \rightarrow 2 \rightarrow 4 \rightarrow 3$.

## 4.3 Flood Reponse

### 4.3.1 Frequency and Magnitude

The total proportion of flood events in neurons 1 to 4 is equal to 35%, 34%, 20% and 11%, respectively. The frequency of floods in each neuron for the streamflow gauges analyzed here is shown in Fig. 9. Neurons 1 and 2 dominate most floods across UPRB. Neuron 3 dominates the floods along the gauges located in the Paranapanema sub-basin (see Fig. 15), while neuron 4 is most associated with floods in the gauges along the Paraná river, particularly with the Itaipu gauge located in the basin outlet,
which interestingly is not directly affected by the wave activity of neuron 1 (see following discussion).

The magnitude of floods associated with each neuron class is analyzed by calculating, for each site, the empirical exceedance probability for each data point in the partial duration series, aggregating all estimates across the sites and then estimating the density of such probabilities conditional on the neuron class of the data points. The results are shown in Fig. 10. Neurons 1 and 2 have the peak and largest density in small values of exceedance probability, suggesting that the biggest floods along UPRB
are associated with these patterns of rainfall (Fig. 3) and moisture transport and convergence (Fig. 4). It is worth mentioning that neuron 2 has a rainfall dynamics that is not associated with El Niño events (Fig. 8), but still produces large floods. The pattern of neuron 3 is more associated with intermediate magnitude flood events while neuron 4 is remarkably associated with the smallest flood events, although some large flood events are possible, particularly in the sites where this neuron dominates the frequency of occurrence (Fig. 9).

### 4.3.2 Spatial Scaling

The literature of the scale of flood properties (e.g. quantiles) with drainage area (Farquharson et al. (1992); Gupta and Dawdy (1995); Gupta et al. (1994, 2007); Gupta and Waymire (1990); Over and Gupta (1994); Pandey et al. (1998)) suggest that the type of precipitation (e.g. convective versus frontal) and the attributes of the drainage network will jointly determine the different behaviors of the scaling process of flow and drainage area. It is not clear whether such scaling relations will hold,
if a mixture of mechanisms can interact to produce large floods. Here we explore the scaling of the first and second sample moments of the flood events with respect to the neuron classes.



Since each flood event at a given site can be assigned to a neuron class, we can easily calculate the sample moments (mean and variance in our case) in each neuron class for each gauge and evaluate how the scaling law of flow moments and drainage area change as a function of the spatio-temporal variability of the rainfall field. Figure 11 shows the scaling of the average flow and drainage area for each neuron class. The magnitudes of the slope and intercept coefficients clearly change as a function of

the neuron class, but more remarkable differences appear between neurons 1/2 and neurons 3/4. In fact, both slope and intercept estimates of either neurons 1 or 2 are significantly different at the 5% significance level from the estimates for neurons 3 and 4.

The magnitude of these coefficients also reflects the rainfall pattern associated with each neuron (Fig. 3). As the rainfall intensity increases, it is expected that the intercept will increase, while the slope is more related to the spatial homogeneity of the rainfall field: as it becomes more homogeneous across the basin, we expect the slope will approach 1. The intercept

values as shown in Fig. 11 increase from neuron 4 to neuron 1, which qualitatively agrees with the rainfall patterns showed in Fig. 3, whose overall magnitude increases from neuron 4 to 1. The slope estimates suggest that the less homogeneous rainfall fields occur in neurons 1 and 2, which is consistent with the pattern displayed in Fig. 3. Neurons 3 and 4 have the largest slope estimates and thus more homogeneous rainfall field, which is again consistent with the results of Fig. 3.

The scaling of the sample variance with the drainage area for each neuron class is shown in Fig. 12. As for the average

flow scaling, the largest differences among the coefficients are observed between the pair of neurons 1 and 2 and the pair of neurons 3 and 4. Visually, the scaling is clearer for neurons 1 and 2. Neuron 4 shows more dispersed values along the least squares regression line, suggesting that the mechanisms by which this rainfall pattern produces a given flood across the gauges, particularly for small gauges, are different (see subsequent discussion).

### 4.3.3   Flood Event Synchronization

The understanding of how each neuron produces a given spatial dynamics of floods across UPRB will be qualitatively explored here through the concepts of event synchronization and complex networks, which have been successfully applied in many fields (Quiroga et al. (2002)) and also climate science (Marwan and Kurths (2015); Malik et al. (2012)), including for prediction of floods in South America (Boers et al. (2014)). Following the nomenclature of Quiroga et al. (2002), let us define the time series of flood event dates (obtained from the partial duration series) for two given streamflow sites $x$ and $y$ as $t_i^x$ and $t_j^y$, where

$i = 1, \ldots, m_x$ and $j = 1, \ldots, m_y$. We define two *synchronous* flood events whenever the distance between $t_i^x$ and $t_j^y$ is less than a given time lag $\tau$. Let then $c^\tau(x|y)$ be the number of time in which a flood event in $x$ follows, within the time lag $\tau$, a flood event in $y$:

$$c^\tau(x|y) = \sum_{i=1}^{m_x} \sum_{j=1}^{m_y} J_{ij}^\tau \tag{7}$$





where

$$J_{ij}^{\tau} = \begin{cases} 1 & \text{if} & 0 < t_i^x - t_j^y \leq \tau \\ 1/2 & \text{if} & t_i^x = t_j^y \\ 0 & \text{otherwise.} \end{cases} \qquad (8)$$

Similarly, we can calculate $c^{\tau}(y|x)$. We will define then a measure for the event synchronization:

$$Q_{ij} = \frac{c^{\tau}(x|y) + c^{\tau}(y|x)}{\sqrt{m_x \cdot m_y}}, \qquad (9)$$

where $0 \leq Q_{\tau} \leq 1$, and $Q_{\tau} = 1$ suggest fully synchronization.

The delay behavior (or *direction of flow*) of the flood events can be measured by:

$$q_{ij} = \frac{c^{\tau}(x|y) - c^{\tau}(y|x)}{\sqrt{m_x \cdot m_y}}, \qquad (10)$$

where $-1 \leq q_{\tau} \leq 1$, and $q_{\tau} = 1$ implies that flood events in $x$ always precede flood events in $y$.

When combining all streamflow sites, $Q_{ij}$ will be a square symmetric matrix while $q_{ij}$ will be a square antisymmetric matrix. $Q_{ij}$ can then be converted into a square binary matrix where entries will represent only relevant connected sites. This can be accomplished by constructing the adjacency matrix $\mathbf{A}$:

$$\mathbf{A} == \begin{cases} 1 & \text{if} & Q_{ij} > T \\ 0 & \text{otherwise,} \end{cases} \qquad (11)$$

where $T$ is a given threshold.

Methods to estimate $T$ usually involve a bootstrap procedure, so that only a certain percentage of the total number of grid points (e.g. 5%) are connected (Boers et al. (2014); Malik et al. (2012)). In the particular case of this work, we are more interested, for a given gauge, in the most gauges that have somehow synchronized flood events with it. Hence, we will define $T = 0.5$, so that we define synchronized gauges when at least 50% of their flood events occur simultaneously.

The time lag $\tau$ should be less than half the minimum inter-event distance, so that one single flood event is not synchronized with two events in another site. Based on this, a simple mathematical formulation is presented in Quiroga et al. (2002). In our case, in order to consider independent flood events, we have defined the partial duration series so that flood events are at least 15 days apart. Hence, $\tau = 7$. The average direction in which the flood event propagates will be simply evaluated by the sign of $q_{ij}$.

Figure 13 shows a directed network obtained from the adjacency matrix $\mathbf{A}$ and the delay behavior matrix $q_{ij}$ considering all flood events across sites and not taking into account the neuron classes. The nodes represent the streamflow gauges in





their geographical position while the edges the existence of synchronization between two sites. The arrow shows the dominant direction of the flood propagation. The flow patterns generally follow the drainage basin direction (Fig. 1): east-west and north-south. But some exceptions can also be observed, indicating that the size and movement of the storm path may also affect how the sites are synchronized.

If we cluster the flood events into the neuron classes, we can obtain specific adjacency and delay behavior matrices for each neuron. The resulting directed networks are shown in Fig. 14. Now we can observe that the rainfall pattern described by neuron 1 produces the largest synchronization of flood events, including inter- and intra-subbasins connectivity. In general, the cascade of flood events tend to end up in the outlet of the sub-basins (see Fig. 15 for the name and location of the sub-basins). Neuron 2 has a more intra-subbasin connectivity pattern, that tends to follow the river flow direction and suggest that rainfall upstream

of the basin is the more likely cause of floods. The Itaipu site located in the basin outlet is not connected to any site, suggesting that Itaipu floods in this neuron will likely result from the routing flow from upstream sites. Neuron 3 has the northern sites disconnected while a connectivity within and across sub-basins is observed. The Tietê subbasin seems to be disconnected from all other subbasins. Finally, neuron 4 show less connections, were most of them are within the subbasins. The Itaipu site is again completely disconnected, so most of its floods associated with neuron 4 are due to routing of upstream flow and floods

caused by rainfall of this and other types.

## 5   Summary and Conclusions

A general, statistical approach to classify flood generation mechanisms, the areal scaling of floods, and the synchronization potential of flooding in a large river basin, was developed and exemplified with data from the Upper Paraná River Basin, Brazil. This is the first attempt to describe such floods in a broad, hydroclimate context. A Self-Organizing Map algorithm

was employed to find the spatio-temporal dynamics of the rainfall field over the basin in the days that preceded the major flood events. For each cluster, we analyzed the large scale moisture transport into the region as well the upper level structure and teleconnections associated with SST. The flood response associated with each rainfall pattern was evaluated in terms of magnitude, frequency, spatial scaling and events synchronization.

Four distinct patterns of rainfall were observed and associated with the atmospheric circulation and moisture transport. The

first cluster exhibits strong rainfall concentrated in the northeastern part of the basin, with a peak two days before the flood events. It was associated with the moisture transport from the Amazon and intense SACZ, with the presence of cyclones - a pattern that have also been reported in the literature (Liebmann et al. (1999); Robertson and Mechoso (2000)). These events are associated with positive SST anomalies in the tropical Pacific and Atlantic oceans and a dipole structure off the eastern coast of South America, which has also been observed in other studies (e.g. Doyle and Barros (2002). On average, 35% of all

floods happen during these conditions. The Itaipu streamflow gauge located in the basin outlet is less affected, at least directly, by this rainfall pattern. These type of floods are strongly synchronized across all sites.

The third neuron shows features of SACZ episodes associated with extratropical disturbances, possibly fronts and cyclones. The rainfall field is however less intense than that of neuron 1 and peak in the southwestern part of the basin. The composite



analysis for the SST field has a pattern similar to that of neuron 1, but the seesaw structure off the eastern South America coast is reverse. On average, 20% of the floods happen to occur in neuron 3, but this frequency is larger for sites located in the southern part of the basin, particularly in the Paranapanema subbasin. The magnitude of these type of floods are intermediate and there is a synchronization intra- and across the central and southern subbasins, suggesting connectivity due to the storm

track extension and movement and the flood routing along the stream channels. Both neurons 1 and 3 have positive SST anomalies in the ENSO region, which has been also associated with extreme rainfall events in the region (Camilloni and Barros (2003); Grimm and Tedeschi (2009); Cavalcanti et al. (2015)). Neuron 2 has a rainfall peak in the northeastern part of the basin, between 4 and 5 days before the flood event. The average rainfall field is less intense than neuron 1 but more intense than neuron 3.

The moisture path shows warm and moist meridional flow across the entire basin, resulting in rainfall possibly due to low level convergence or eventually frontal activity. The SST field in the Atlantic ocean is similar to that of neuron 1, but the average conditions in the Tropical Pacific are neutral. On average, 34% of floods are of this type, particularly in the northern subbasins. Together with floods in neuron 1, these are the largest floods in the region. The synchronization of type 3 floods are more intra-subbasins. Finally, type 4 floods are caused by an homogeneous but persistent rainfall field, with most moisture

transported from the Atlantic ocean. There is no evidence of directly extratropical activities and the SST field revealed negative anomalies in the tropical Pacific and positive in the tropical Atlantic. The near surface air temperature in this cluster showed positive anomalies, suggesting that local convection might be also an important factor. 11% of the total floods are of these type, although this is the dominant pattern of rainfall. These are the less intense floods, with a synchronization that occurs along the main river channels.

The spatial scaling exponents of floods with drainage area are similar for floods of types 1 and 2, and for types 3 and 4, even though the rainfall mechanisms are different for each pair. The exponent is higher for types 3 and 4 reflecting the higher homogeneity in the rainfall and response pattern. The area exponents for flood variance are considerably higher than those for mean scaling, opening the possibility of a multi-scaling approach. However, once again the exponents are similar for types 1 and 2, and for types 3 and 4. The scaling relationships for variance are not as well constrained for neurons 3 and 4 types of

events.

Distinct patterns of flood synchronization and movement are also identified for each neuron. Conditional on the storm track, i.e., large scale atmospheric flow, these could be further useful to improve analysis and prediction of the potential flood emergence and for the operation of multi-stage flood control systems.

The results obtained in this work are a step forward to improve the flood risk management in UPRB in two possible ways:

flood design and short term prediction. Local flood frequency analysis could make use of the different flood categories and employ, for instance, mixture of distributions approaches (e.g. Alila and Mtiraoui (2002)) for a better flood quantile estimates. Regional flood frequency analysis could also consider the different scaling laws and develop a Bayesian approach (as in Lima and Lall (2010); Lima et al. (2016)) to better estimate regional parameters. Finally, the persistent regions with SST anomalies could be used to derive climate predictors for short term flood risk prediction. The synchronization of the flood

events could be explored in more details to develop short term flood forecast models conditional on the atmospheric and oceans



states and flood situation in nearby sites. Further details of the moisture transport and high level atmospheric circulation could be also analyzed in order to obtain potential climate predictors for the floods in this region. Other attributes of the distributions associated with each flood type were not explored here and will be theme of future work. The timing of the floods along the warm season and a possible association with the neuron classes can be further explored too. As future research, we intent to

5    address part of these topics and also explore how the tools and methodology employed in this work could help evaluate the future flood risk in the UPRB region considering climate changes.

## 6   Data availability

The streamflow data for the Upper Paraná River Basin are provided by the Brazilian National Operator of the System (ONS) and can be accessed at http://www.ons.org.br/home/. The rainfall and temperature data are provided by Xavier et al. (2016) and

10   can be accessed at http://careyking.com/data-download/. The ERA Interim global data set (SST, moisture fluxes, divergence field, vorticity, wind field) are available at http://apps.ecmwf.int/datasets/data/interim-full-moda/levtype=sfc/.

*Competing interests.*   The authors declare that they have no conflict of interest.

*Acknowledgements.*   We thank all agencies and authors that provide dataset and codes. The first author acknowledges a Postdoctoral Fellowship from the Brazilian Government Agency CNPq during part of this work.



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





**Figure 1.** The Parana River Basin (red contour) and streamflow gauges used in this work (black dots). The elevation is in meters and the location of the Parana River Basin within Brazil is showed in the insert in the upper right corner (red line contour).





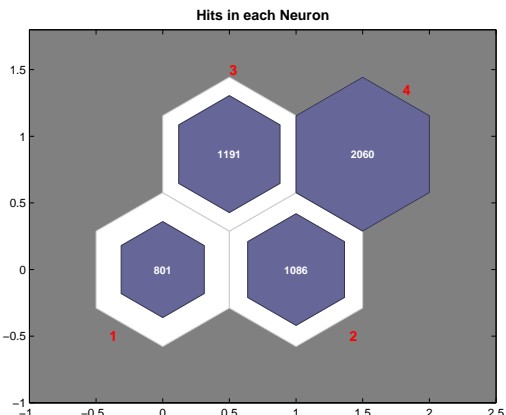
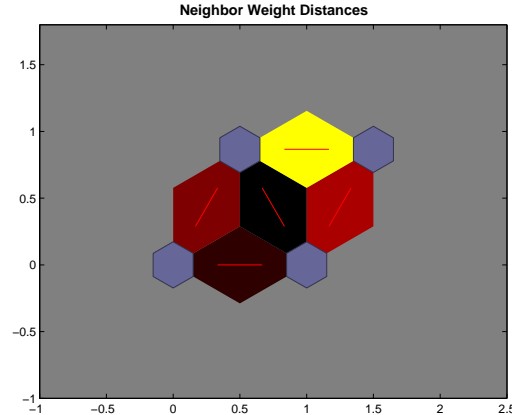

**Figure 2.** Left panel: number of hits in each neuron ( blue hexagons). Right panel: connecting neighboring neurons (red lines). The colors in the regions containing the red lines indicate the distances between neurons, where darker colors represent larger distances and lighter colors represent smaller distances.







**Figure 3.** Neuron weights obtained for the Self-Organizing Map. These weights basically represent the rainfall anomalies (in mm) over the Upper Paraná River Basin from day $t - 5$ (top panels) to day $t$ (bottom panels). The black line shows the zero contour.



**Figure 4.** Vertically integrated moisture fluxes (in kg/(m · s)) and associated divergence field (in $10^{-5}$ kg/(m$^2$· s)) averaged over each neuron class. The red contour line shows the Upper Paraná River Basin. The contour for the divergence field equals to zero is shown by the black lines. The blue contour line shows South America.





**Figure 5.** Streamlines for the vertically integrated moisture fluxes and low level (850 mb) relative vorticity (in $10^{-5} \cdot$ 1/s) averaged over each neuron class. The red contour line shows the Upper Paraná River Basin. The blue contour line shows South America.





**Figure 6.** Streamlines for the high level (500 mb) wind vector and relative vorticity (in $10^{-5} \cdot$ 1/s) averaged over each neuron class. The red contour line shows the Upper Paraná River Basin. The blue contour line shows South America.



**Figure 7.** Anomalies (in °C) in the near surface air temperature averaged over each neuron class. The red contour line shows the Upper Paraná River Basin. The black line shows the zero contour.





**Figure 8.** 15-day filtered sea surface temperature (SST) anomalies (in C°) averaged over each neuron class.





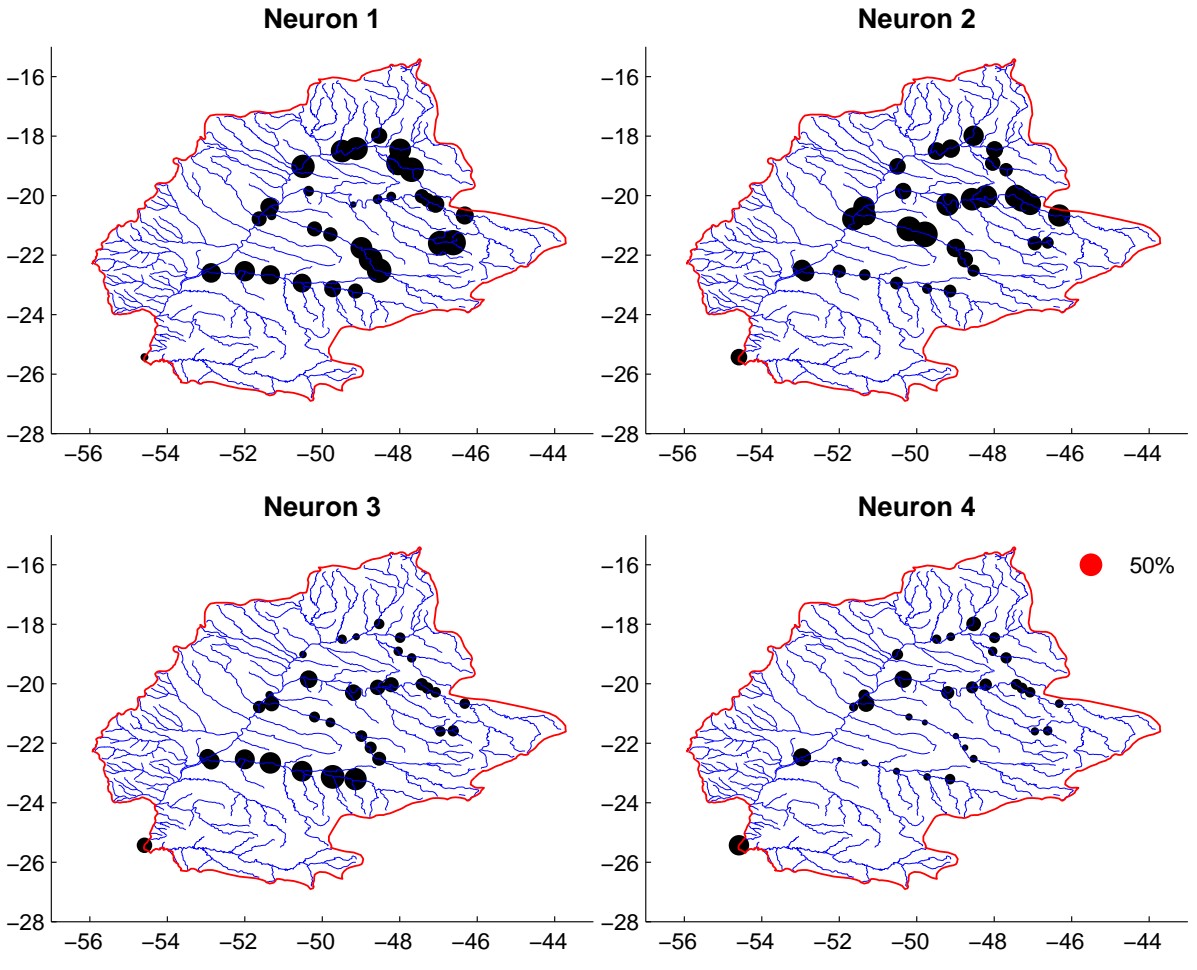

**Figure 9.** Frequency of flood events in each neuron class for each streamflow gauge. The red dot shows the scale for a frequency of 50%.

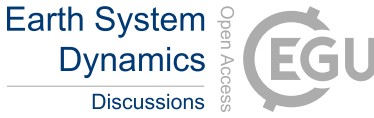



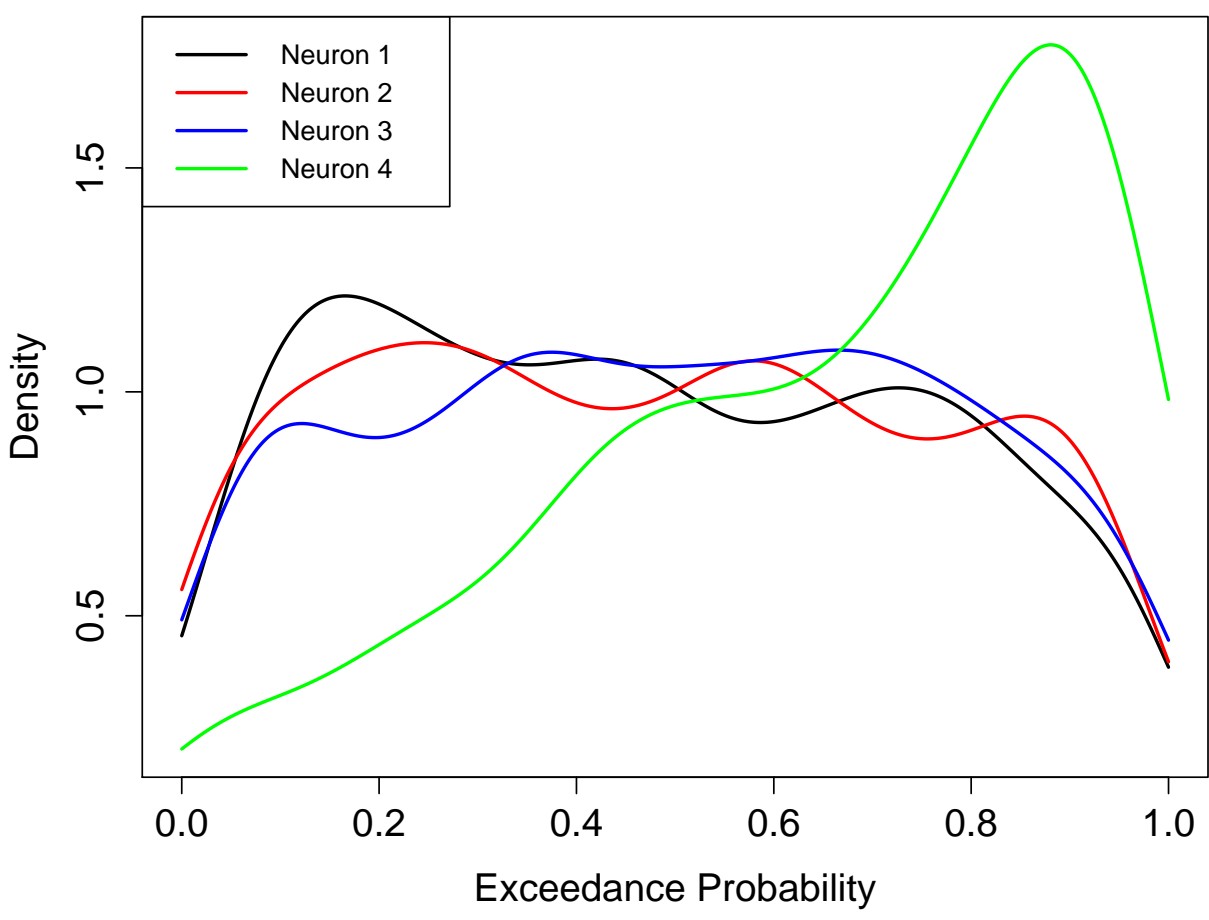

**Figure 10.** Density of exceedance probabilities in each neuron class.







**Figure 11.** Scaling of average flood flow series in each neuron class. The least square estimates of intercept and slope are shown in each panel. The black line shows the least squares regression.


**Figure 12.** Scaling of variance of flood flow series in each neuron class. The least square estimates of intercept and slope are shown in each panel. The black line shows the least squares regression.




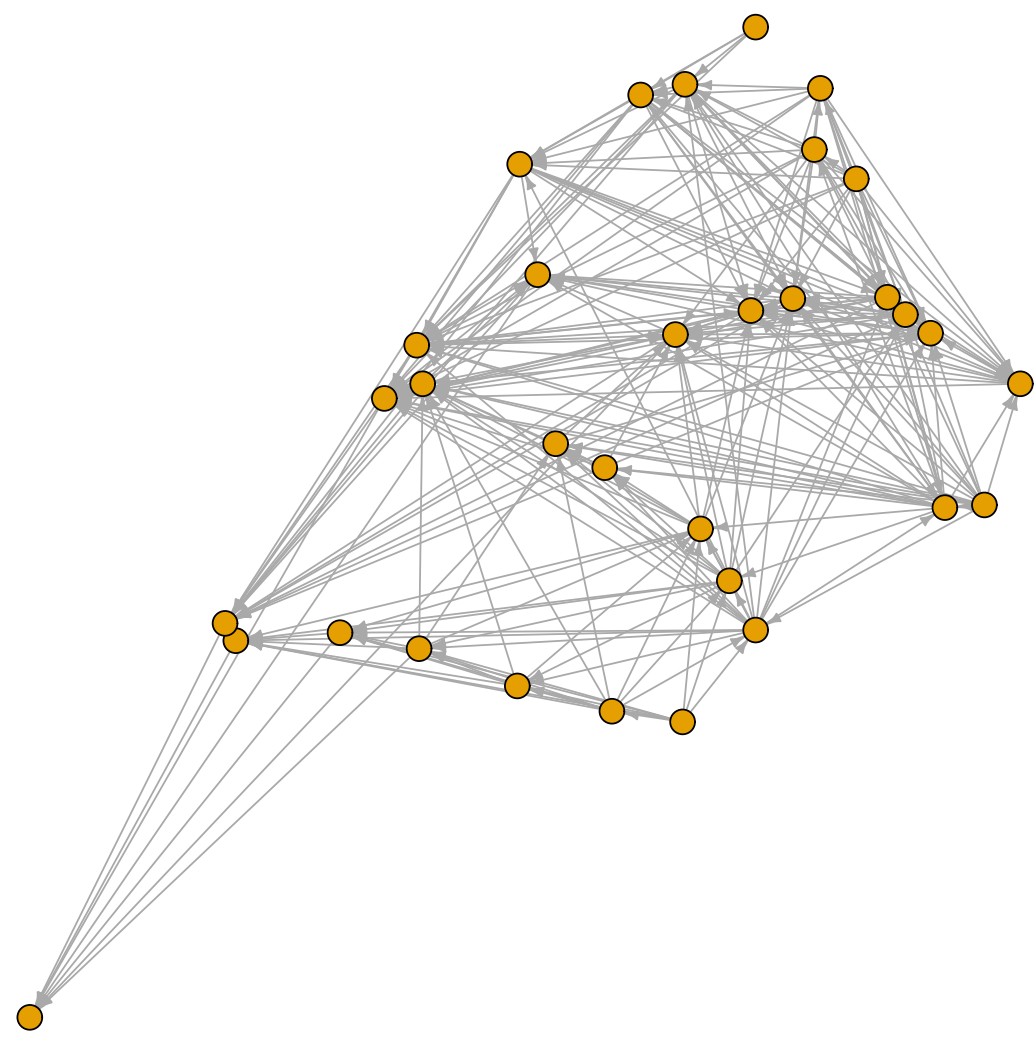

**Figure 13.** A directed network for the flood events showing synchronization and flow direction (arrows). The dots show the streamflow gauges in their geographical location (see Figure 15).





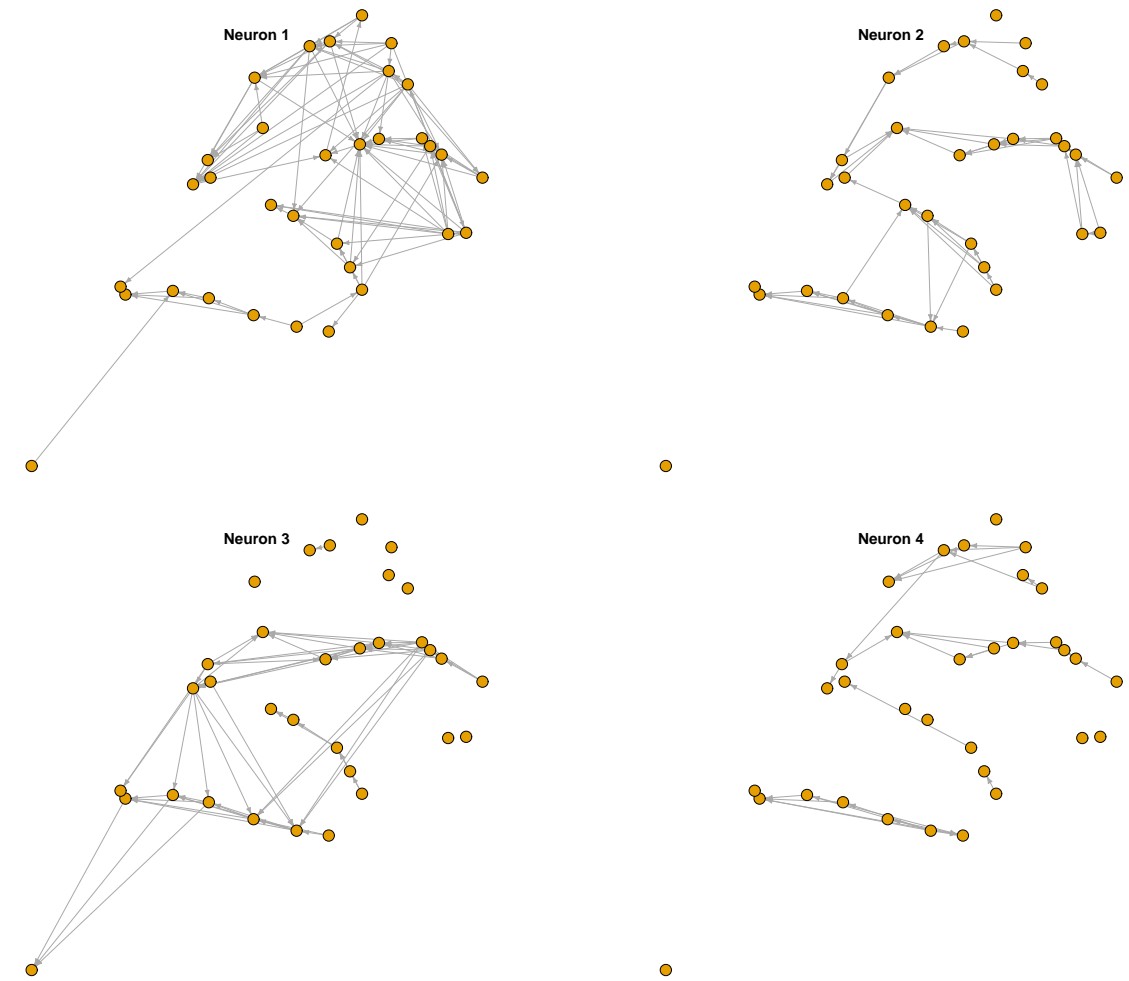

**Figure 14.** A directed network for the flood events showing synchronization and flow direction (arrows) as a function of neuron class. The dots show the streamflow gauges in their geographical location (see Figure 15).



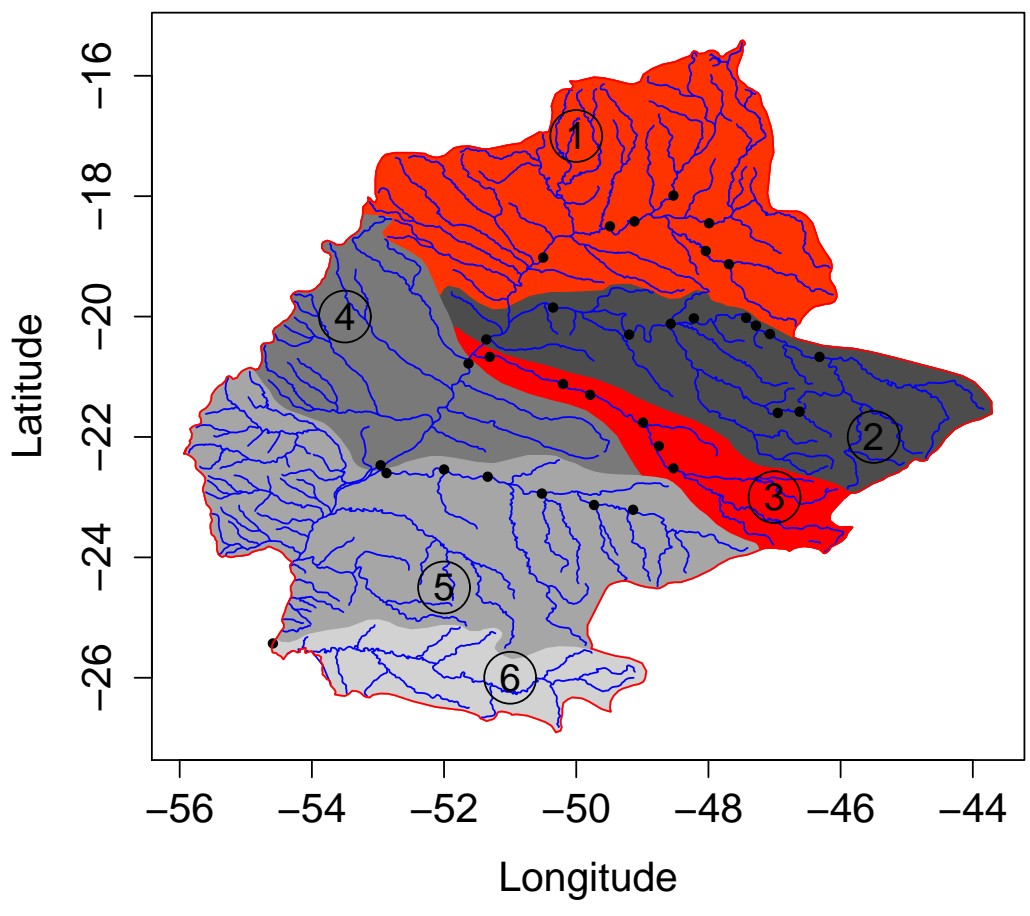

**Figure 15.** Streamflow gauges and associated subbasins: 1) Paranaíba; 2) Grande; 3) Tietê ; 4) Paraná ; 5) Paranapanema/Paraná and 6) Iguaçu.



**Table 1.** Transition probabilities among neurons

| To neuron<br>From neuron | 1 | 2 | 3 | 4 |
|---|---|---|---|---|
| 1 | 0.631 | 0.347 | 0.020 | 0.003 |
| 2 | 0.045 | 0.621 | 0.117 | 0.217 |
| 3 | 0.172 | 0.067 | 0.690 | 0.071 |
| 4 | 0.020 | 0.026 | 0.110 | 0.843 |