# Peer review of "Classification of mechanisms, Climatic Context, Areal Scaling, and Synchronization of floods: the hydroclimatology of floods in the Upper Paraná River Basin, Brazil"

_Earth System Dynamics, 2017_

## Referee Comment (RC1) · E. S. P. R. Martins (Referee) · 8 Apr 2017

E. S. P. R. Martins (Referee)

espr.martins@gmail.com

Classification of mechanisms, Climatic Context, Areal Scaling, and Synchronization of floods: the hydroclimatology of floods in the Upper Paraná River Basin, Brazil

by Carlos Lima, Amir AghaKouchak, and Upmanu Lall

In my view, the authors' contributions are relevant to the field and of interest of those working on improving flood risk management either in flood design or in short term prediction. It is not of my knowledgement that this paper was published elsewhere.

1. Does the paper address relevant scientific questions within the scope of ESD? In my view, yes. The main contribution of the paper is establish a link between flood frequency and flood generating mechanisms. The authors assume the Hirschboeck's hypothesis that "exceptional floods in basis of all sizes could be related to anomalies in the large scale atmospheric circulation" (Hirschboeck, 1988). Such approach has not been applied in South America basins.

2. Does the paper present novel concepts, ideas, tools, or data? The authors propose a statistical approach in order to classify flood generation mechanisms, spatial scaling of floods, and flood event synchronization in a large river basin. This was exemplified with data from a basin located in Brazil.

3. Are substantial conclusions reached? In my view, yes. Specifically:

a. Four distinct patterns of rainfall were observed and associated with the atmospheric circulation and moisture transport.

b. Associated with these patterns, the authors identified also four types of floods for the analyzed basin.

c. It was also identified that the spatial scaling exponents of floods as a function of drainage area are similar for floods types 1 and 2, and for types 3 and 4. The exponent is higher for types 3 and 4 than those for floods types 1 and 2. The area exponents for flood variance are considerably higher than those for mean scaling, which, according with the authors (and I agree), points out to the possibility of a multi-scaling approach.

d. The techniques used were also able to identify distinct patterns of flood synchronization and movement, which were conditional to the sorm track. This has a potential use to improve analysis and prediction for flood emergency and flood control systems purposes.

4. Are the scientific methods and assumptions valid and clearly outlined? Yes, the methods and assumptions were clearly outlined. The authors were quite careful in the

mathematical development.

5. Are the results sufficient to support the interpretations and conclusions? Yes, In my view the results provide support to the interpretations and conclusions drawn by the authors.

6. Is the description of experiments and calculations sufficiently complete and precise to allow their reproduction by fellow scientists (traceability of results)? Yes, in my view the work can be reproduced by others scientists – of course, given that these are skillfull in the techniques and methods employed in the manuscript.

7. Do the authors give proper credit to related work and clearly indicate their own new/original contribution? Yes, the authors did an extent review of literature providing the due credit and indicating their own contributions. The authors dia a good job in putting their work in the context of recent literature.

8. Does the title clearly reflect the contents of the paper? In my view, yes.

9. Does the abstract provide a concise and complete summary? In my view, yes.

10. Is the overall presentation well structured and clear? Yes, in general the paper is well written and structured.

11. Is the language fluent and precise? Yes, but there is still room for improvement. I would recommend a last review for english style.

12. Are mathematical formulae, symbols, abbreviations, and units correctly defined and used? Yes.

13. Should any parts of the paper (text, formulae, figures, tables) be clarified, reduced, combined, or eliminated? In my view all the material presented are needed and clear. The paper should not be shortened.

14. Are the number and quality of references appropriate? Yes. The authors, as I already mentioned before, made a good job putting their work into perspective of the

current literature.

15. Is the amount and quality of supplementary material appropriate? Yes, in my view the presented material is enough and of high quality, and it allows the reader to understand fully the methods and the analysis that was made.

---

## Referee Comment (RC2) · S. Harrigan (Referee) · 17 Apr 2017

**Review of Lima et al. (2017) in ESD by Shaun Harrigan**

**A.) General Comments**

Lima et al. (2017) presents a methodological framework, based on self-organising maps (SOM) and composite analysis, for identifying the rainfall and large-scale climatic patterns linked to floods using the Upper Paraná River Basin (UPRB) in Brazil over the 1980-2013 period as a case study. Four primary flood-generating rainfall clusters were identified from the SOM analysis along with large-scale climate (moisture, wind and sea surface temperature etc.) conditions during observed flood events, for each of the four clusters. This paper uncovered interesting new insights into the flood hydroclimatology in UPRB, beyond the simplistic 'El Niño is responsible for all floods' hypothesis, with potential for this new hydroclimatic knowledge to be used to improve flood frequency analysis and flood forecasting. There are however a number of places in which I think this manuscript could be improved to best get across key messages as highlighted in my specific comments in section B. Overall, the layout of the manuscript, in my opinion, does not do the work justice. I've made more specific points below, but in several places some methods are mixed in with results and there is no distinct discussion section. There are many different steps within the framework but they all rely heavily on the initial SOM analysis, I outline several methodological points of clarification for the authors. The overall presentation of the paper would be greatly improved with an increased level of copy-editing both in-terms of language and figures. I support publication of this paper in the ESD special issue, in principle, and hope the authors can spend the time to tighten and clarify their approach as it would make a valuable contribution to the hydroclimatogical literature.

**B.) Specific Comments**
1. Glad to see Hirschboeck (1988) being cited as shows the field of hydrolclimatology has some history, although it is only relatively recently that the benefit of the hydroclimatic perspective is being fully appreciated – this paper is therefore a welcome addition to the growing literature on hydroclimatology. As general point of interest (not required to include), the first definition of hydroclimatology I found was by Langbein (1967).

2. You mention in the abstract (Pg1; L6-9) that a Eulerian-Lagrangian model of ocean-atmosphere circulation would ideally be needed…", "However, some progress may be possible through empirical data analysis.". I agree with you here but this point needs to be raised in the introduction and expanded. What is the benefit of the empirical analysis, what progress can be made, what is the justification of this approach over others/is it complementary to other approaches?

3. Along the lines of the above point, you base a lot of the results on the Self-Organizing Maps (SOM) analysis. I have no issue with the use of SOMs, and commend the authors for a rigorous application of the method, however there is little justification of why this method was chosen over others? What particular advantages does SOMs provide in comparison to other more widely used classical methods of classification and clustering (e.g. PCA, K-means clustering, etc.)?

4. Pg4; L21: To avoid confusion for the international audience I would recommend referring to the flood season as the 'wet season' too rather than just 'warm season' throughout the manuscript. Also, is there an approximate % of total floods that occur in Nov-March (i.e., > 60% or > 95%) rather than just stating "most"?

5. Pg4; L22-27: Peak-Over-Threshold (or partial duration series) – The extraction of a POT series from daily flow can be challenging, especially for more groundwater influenced catchments

with longer memory, hence the need for your independence/declustering criteria. Some literature to support your decision of 15 days could be Mallakpour and Villarini (2015b) or Svensson et al. (2005). On average you have extracted about a POT3 series (i.e. on average ~3 events per year [98 peaks over 34 year record] or POT2.88 to be more precise). However, can you give a range across the 33 basins if there are large deviations from the 98 event average (i.e. I would expect fewer independent peaks to be extracted from more groundwater catchments, and more from flashier headwater catchments) using a fixed threshold as used here.

6. Pg4; last para: What is the grid resolution of the rainfall dataset?

7. Pg5; L9: Is it only climate/SST datasets that are interpolated to 2.5 deg grids, or is rainfall also interpolated? Why is such a coarse resolution used given ERA-Interim is at 0.75 deg resolution?

8. Pg8; L3-13: This is methods description and should be moved to previous methods section, rather in the results section.

9. Pg8; L3-4: I'm not an expert in SOMs so forgive my ignorance, however, your decision to decide on K = 4 clusters appears arbitrary and given it defines everything thereafter (e.g. text from the Abstract: "classify […] UPRB into four categories" and "classify floods into four types".). There needs to be some physical basis/justification to guide this decision.

10. Pg8; L30-31: By concluding cluster 4 reflects average rainfall conditions during the rainy season, you're essentially implying that the largest basin(s) in UPRB flood under average rainfall conditions (i.e. Figure 9 Neuron 4 panel). An alternative explanation could be that the larger basin reports a flood only when rainfall conditions are generally 'wet'/'moderately wet' for long periods of time (perhaps with wet antecedent conditions much longer than 5 days) and over the entire basin, rather than from more localised rainfall anomalies as is the case in clusters 1-3. This is also reflected in the fact the transition probability of Neuron 4 to Neuron 4 is highest (0.843 in Table 1).

11. Understanding rainfall/flood clusters: Are the 5138 days of rainfall data as used to identify the four rainfall clusters (i.e. from Pg7; L27-28) divided evenly (i.e. ~1285 days contributing to each cluster)? Following this, is the composite analysis (Pg9; L8-11) conducted only on the sub-selection of those rainfall days within each cluster that also had a reported flood event across the 33 river basins over the 1980-2013 period? If this is so, how many days within each cluster contribute to the composite analysis?

I'm getting slightly confused here as you state that all 5138 rainfall days are used as input to SOM (Pg7; L27-28) then mention on Pg14; L20-21 that SOM was employed to find dynamics of "the rainfall field over the basin in the days that preceded the major flood events". Can you clarify this step for me please?

12. Pg10; L20: Need to be more specific when discussing 'El Niño region' (i.e. Niño 3, 3.4, or 4?) or make clear it's the broader area you are talking about – some people get very picky when it comes to ENSO definitions!

13. Pg11; L8: Should neuron 1 feature in sequence of neuron transitions?

14. Pg11; L20-21: The point about large floods being generated under non-El Niño conditions is an important one that should be discussed more in the context of the wider international

literature (You introduce such papers that do state El Niño-flood links in UPRB on Pg4; L11-13 but don't discuss your results again in this wider context). It is often assumed, wrongly, that majority/all flooding in South America is due to El Niño, this work suggests in UPRB things are more complex and uncertainties exist (also see Emerton et al., (2017)).

**15.** Pg15; L5-6: Could you make a tentative conclusion that about 55% of floods (i.e. 35% in neuron 1 + 20% in neuron 3) are linked to El Niño-like SST patterns? I also acknowledge that El Niño events have more strict definitions regarding strength and persistence of positive SST anomalies in a fixed region of the Pacific.

**16.** Following the above point, the SST pattern in neuron 4 (Fig. 8) is similar to La Niña-like conditions with negative SST anomalies, so could the 11% of floods under neuron 4 be linked with this large-scale phenomenon, even tentatively? If further analysis suggests so, then it is interesting to conclude that floods in UPRB can occur under both El Niño and La Niña conditions.

**17.** Layout of paper: It is my opinion that the Impact of the paper would be greater if the layout was slightly modified – there is currently no distinct discussion section and some methods descriptions are mixed in with the results section (e.g. Pg8; L3-13). Renaming Section 4 to 'Results and Discussion' (or having a dedicated 'Discussion' section) and move some of the discussion from Section 5 (currently 'Summary and Conclusions') to Section 4, and rename Section 5 to 'Conclusions' would be my suggestion.

**18.** Comment on Figures: The foundation of this paper (and indeed the SOM method) is on visual display of results on maps. Figures 3-8 relay on the 'rainbow' colour scheme that makes distinguishing patterns difficult – a divergent colour scheme that had a neutral (or while) colour for zero values with diverging colours for positive and negative anomalies would be much more effective. I do note the authors include the zero contour line, but this is still misleading in places. The "end the rainbow" calls are well known and with good scientific basis (Light and Bartlein (2004) and this 2014 post by Ed Hawkins et al. https://www.climate-lab-book.ac.uk/2014/end-of-the-rainbow/). I can only make this a suggestion for improvement but it's ultimately up to the authors/journal as many papers/journals are still using this colour scheme.

**19.** Further refinement of figure axes and more descriptive captions would be beneficial: For example, what are the units (if any) in Figure 2; adding "t-5, t-4, … t" to the y-axis in Figure 3 would be more visually impactful.

**20.** Figure 15 could be combined within Figure 1.

**C.) Technical Corrections**

Mostly well written but would benefit from a final proof read to tidy-up grammar – (i.e. abstract and summary and conclusions, in particular, tense in Section 3 should be in past tense).

Some things to help cleaning up:

**21.** Accent on 'Paraná' is not used consistently throughout
**22.** In-text references to be in correct format
**23.** Pg 2; L29: change 'basis' to 'basin'

24. Pg2; L34: It is a huge claim that there are no studies whatsoever on the broad topic of flood hydroclimatology in South America. It would be more appropriate to state something like "However, there is a lack of knowledge on the flood hydroclimatology of South America.".
25. P3; L28: Change 'Dataset' to 'Datasets'
26. Pg4; L30: Acronyms not defined
27. Pg5; L28: Change 'will adopt' to 'applied'
28. Reference for Merz et al., (2014) is for discussion paper and not final published

**References**

Emerton, R., Cloke, H. L., Stephens, E. M., Zsoter, E., Woolnough, S. J. and Pappenberger, F.: Complex picture for likelihood of ENSO-driven flood hazard, Nat. Commun., 8, 14796, doi:10.1038/ncomms14796, 2017.

Langbein, W. G.: Hydroclimate. In Fairbridge, R.W., ed., The Encyclopedia of Atmospheric Sciences and Astrogeology, New York, Reinhold, pp. 447–451, 1967.

Light, A. and Bartlein, P. J.: The end of the rainbow? Color schemes for improved data graphics, Eos Trans. Am. Geophys. Union, 85(40), 385–391, doi:10.1029/2004EO400002, 2004.

Mallakpour, I. and Villarini, G.: The changing nature of flooding across the central United States, Nat. Clim. Change, advance online publication, doi:10.1038/nclimate2516, 2015b.

Merz, B., Aerts, J., Arnbjerg-Nielsen, K., Baldi, M., Becker, A., Bichet, A., Blöschl, G., Bouwer, L. M., Brauer, A., Cioffi, F., Delgado, J. M., Gocht, M., Guzzetti, F., Harrigan, S., Hirschboeck, K., Kilsby, C., Kron, W., Kwon, H.-H., Lall, U., Merz, R., Nissen, K., Salvatti, P., Swierczynski, T., Ulbrich, U., Viglione, A., Ward, P. J., Weiler, M., Wilhelm, B. and Nied, M.: Floods and climate: emerging perspectives for flood risk assessment and management, Nat Hazards Earth Syst Sci, 14(7), 1921–1942, doi:10.5194/nhess-14-1921-2014, 2014.

Svensson, C., Kundzewicz, W. Z. and Maurer, T.: Trend detection in river flow series: 2. Flood and low-flow index series, Hydrol. Sci. J., 50(5), 811–824, doi:10.1623/hysj.2005.50.5.811, 2005.

---

## Referee Comment (RC3) · Anonymous Referee #3 · 7 May 2017

A methodological framework based on the Self-Organizing Map (SOM) clustering was employed to find the spatio-temporal dynamics of the rainfall field over the Upper Paraná River Basin in Brazil in the days that preceded the major flood events. For each cluster, the large-scale moisture transport into the region was analyzed as well the upper level structure and teleconnections associated with sea surface temperature. The flood response associated with each rainfall pattern was evaluated in terms of magnitude, frequency, spatial scaling and events synchronization.

I agree with the authors that the findings and the methodological framework proposed

in this study provide new insights for understanding causes of floods around the world and are a step forward to improve flood risk management and interpreting statistical assessments. I support publication of this manuscript in ESD after minor revisions:

Specific issues:

1. Please provide additional discussion on how the proposed approach help short-term flood forecasting.

2. Please provide assumptions and limitations of the proposed approach, mainly due to SOM approach and using ERA-Interim reanalysis datasets.

3. Clarify "typically of some km2" in line 7 of page 3.

4. Please provide justification or the limitations of using 2.5o x 2.5o resolution ERA-Interim reanalysis datasets (moisture fluxes, vorticity, upper level winds and sea surface temperature) in the proposed approach for investigating the basin scale flood mechanisms.

---

## Author Comment (AC2) · 13 Jun 2017

We would like to thank Dr. Shaun Harrigan for the constructive and thoughtful comments. We will address all the comments in the revised version. Our responses to the main comments raised by Dr. Shaun Harrigan  (_underlined and in italic_) are provided below.

_Glad to see Hirschboeck (1988) being cited as shows the field of hydrolclimatology has some history, although it is only relatively recently that the benefit of the hydroclimatic perspective is being fully appreciated – this paper is therefore a welcome addition to the growing literature on hydroclimatology. As general point of interest (not required to include), the first definition of hydroclimatology I found was by Langbein (1967)._

We agree and we thank the reviewer for pointing us to   by Langbein (1967).

_You mention in the abstract (Pg1; L6-9) that a Eulerian-Lagrangian model of ocean-atmosphere circulation would ideally be needed…", "However, some progress may be possible through empirical data analysis.". I agree with you here but this point needs to be raised in the introduction and expanded. What is the benefit of the empirical analysis, what progress can be made, what is the justification of this approach over others/is it complementary to other approaches?_

We agree and we will extend the discussion. Also, we would like to highlight that we have provided more information about the value and benefits of our approach in lines 1-8 and 18-34 of page 1.

_Along the lines of the above point, you base a lot of the results on the Self-Organizing Maps (SOM) analysis. I have no issue with the use of SOMs, and commend the authors for a rigorous application of the method, however there is little justification of why this method was chosen over others? What particular advantages does SOMs provide in comparison to other more widely used classical methods of classification and clustering (e.g. PCA, K-means clustering, etc.)?_

Note that $\mathbf{X}$ (line 25, p. 5  and lines 25-28, p. 7 ) is a very large matrix (size 5138 x 7068) and, in this case, classical methods of clustering (e.g. K-means, spectral clustering) tend to fail. We have adopted SOMs which has found several applications in climate science (lines 31-34, p. 5) for similar purposes.

_Pg4; L21: To avoid confusion for the international audience I would recommend referring to the flood season as the 'wet season' too rather than just 'warm season' throughout the manuscript. Also, is there an approximate % of total floods that occur in Nov-March (i.e., > 60% or > 95%) rather than just stating "most"?_

We will be happy to make these changes in the revised version of the manuscript and indicate the % of floods that occur in the wet season.

*Pg4; L22-27: Peak-Over-Threshold (or partial duration series) – The extraction of a POT series from daily flow can be challenging, especially for more groundwater influenced catchments with longer memory, hence the need for your independence/declustering criteria. Some literature to support your decision of 15 days could be Mallakpour and Villarini (2015b) or Svensson et al. (2005). On average you have extracted about a POT3 series (i.e. on average ~3 events per year [98 peaks over 34 year record] or POT2.88 to be more precise). However, can you give a range across the 33 basins if there are large deviations from the 98 event average (i.e. I would expect fewer independent peaks to be extracted from more groundwater catchments, and more from flashier headwater catchments) using a fixed threshold as used here.*

We thank the reviewer for pointing us to these references and raising this interesting point. The number of events ranges from 76 to 131, with an average of 98. The reviewer is right: headwater basins have more events than downstream basins. In Figure 1 below we show how the number of floods events increases for headwater basins, which are located in the eastern region of the Upper Paraná basin (please see also Fig. 1 of the manuscript) and have smaller drainage areas.

[Figure]

Figure 1 –Number of floods events versus longitude (left panel) and drainage area (right panel).

*Pg4; last para: What is the grid resolution of the rainfall dataset?*

The grid resolution is $0.25^{o} \times 0.25^{o}$.

*Pg5; L9: Is it only climate/SST datasets that are interpolated to 2.5 deg grids, or is rainfall also interpolated? Why is such a coarse resolution used given ERA-Interim is at 0.75 deg resolution?*

It is only the climate datasets that are interpolated to $2.5^o$. We acknowledge that ERA-Interim is also available at $0.75^o$ resolution, but we think that, for the purposes of this work, the coarser resolution of $2.5^o$ provides satisfactory results.

*Pg8; L3-13: This is methods description and should be moved to previous methods section, rather in the results section.*

We agree with the reviewer and will do this in the revised version of the manuscript.

*Pg8; L3-4: I'm not an expert in SOMs so forgive my ignorance, however, your decision to decide on K = 4 clusters appears arbitrary and given it defines everything thereafter (e.g. text from the Abstract: "classify […] UPRB into four categories" and "classify floods into four types".). There needs to be some physical basis/justification to guide this decision.*

In lines 3-4 of page 8 we provide a justification: we want a relatively large number of flood events in each cluster. Given the four clusters, we obtain, on average, $98/4 \cong 25$ events in each cluster, which we believe can provide some useful flood statistics. We have also noticed that, as we increase the number of clusters for more than 4, similar spatial patterns of rainfall (Figure 3) start to appear in different clusters.

*Pg8; L30-31: By concluding cluster 4 reflects average rainfall conditions during the rainy season, you're essentially implying that the largest basin(s) in UPRB flood under average rainfall conditions (i.e. Figure 9 Neuron 4 panel). An alternative explanation could be that the larger basin reports a flood only when rainfall conditions are generally 'wet'/'moderately wet' for long periods of time (perhaps with wet antecedent conditions much longer than 5 days) and over the entire basin, rather than from more localised rainfall anomalies as is the case in clusters 1-3. This is also reflected in the fact the transition probability of Neuron 4 to Neuron 4 is highest (0.843 in Table 1).*

We offer an explanation for this in page 14, lines 10-11 and 13-15. The Itaipu site is the largest basin in UPRB and we believe that floods there are essentially coming from routing flow from upstream sites. Note also that the rainfall pattern of neuron 4 shows rainfall conditions below the average (negative anomalies) across the entire basin.

*Understanding rainfall/flood clusters: Are the 5138 days of rainfall data as used to identify the four rainfall clusters (i.e. from Pg7; L27-28) divided evenly (i.e. ~1285 days contributing to each cluster)?*

No, in Figure 2 (left panel) we show the number of hits (i.e. days) in each neuron.

*Following this, is the composite analysis (Pg9; L8-11) conducted only on the sub-selection of those rainfall days within each cluster that also had a reported flood event across the 33 river basins over the 1980-2013 period? If this is so, how many days within each cluster contribute to the composite analysis?*

No, the composite analysis is conducted for the total number of days in each cluster.

*I'm getting slightly confused here as you state that all 5138 rainfall days are used as input to SOM (Pg7; L27-28) then mention on Pg14; L20-21 that SOM was employed to find dynamics of "the rainfall field over the basin in the days that preceded the major flood events". Can you clarify this step for me please?*

We use the entire 5138 days as input to SOM. We agree that the sentence on page 14, l. 20-21 is confusing and we will rephrase it in the revised version of the manuscript.

*Pg10; L20: Need to be more specific when discussing 'El Niño region' (i.e. Niño 3, 3.4, or 4?) or make clear it's the broader area you are talking about – some people get very picky when it comes to ENSO definitions!*

We will provide more details of the ENSO region in the revised version.

*Pg11; L8: Should neuron 1 feature in sequence of neuron transitions?*

Yes, it should be 3 → 1 → 2 → 4 → 3. We will correct in the revised version.

*Pg11; L20-21: The point about large floods being generated under non-El Niño conditions is an important one that should be discussed more in the context of the wider international literature (You introduce such papers that do state El Niño-flood links in UPRB on Pg4; L11-13 but don't discuss your results again in this wider context). It is often assumed, wrongly, that majority/all flooding in South America is due to El Niño, this work suggests in UPRB things are more complex and uncertainties exist (also see Emerton et al., (2017)).*

We thank the reviewer to point out this reference. In the revised version, we will extend the discussion and provide a broader perspective. In Section Conclusions (p. 14, lines 24-33 and page 15, lines 1-19) we mention the literature and provide more details about the atmospheric and oceans conditions during flood events.

*Pg15; L5-6: Could you make a tentative conclusion that about 55% of floods (i.e. 35% in neuron 1 + 20% in neuron 3) are linked to El Niño-like SST patterns? I also acknowledge that El Niño*

*events have more strict definitions regarding strength and persistence of positive SST anomalies in a fixed region of the Pacific.*

We will gladly add this to the revised version.

*16. Following the above point, the SST pattern in neuron 4 (Fig. 8) is similar to La Niña-like conditions with negative SST anomalies, so could the 11% of floods under neuron 4 be linked with this large-scale phenomenon, even tentatively? If further analysis suggests so, then it is interesting to conclude that floods in UPRB can occur under both El Niño and La Niña conditions.*

We believe this is an interesting hypothesis that needs further investigation, but is beyond the scope of this paper.

*17. Layout of paper: It is my opinion that the Impact of the paper would be greater if the layout was slightly modified – there is currently no distinct discussion section and some methods descriptions are mixed in with the results section (e.g. Pg8; L3-13). Renaming Section 4 to 'Results and Discussion' (or having a dedicated 'Discussion' section) and move some of the discussion from Section 5 (currently 'Summary and Conclusions') to Section 4, and rename Section 5 to 'Conclusions' would be my suggestion.*

We thank the reviewer´s suggestion.

*18. Comment on Figures: The foundation of this paper (and indeed the SOM method) is on visual display of results on maps. Figures 3-8 relay on the 'rainbow' colour scheme that makes distinguishing patterns difficult – a divergent colour scheme that had a neutral (or while) colour for zero values with diverging colours for positive and negative anomalies would be much more effective. I do note the authors include the zero contour line, but this is still misleading in places. The "end the rainbow" calls are well known and with good scientific basis (Light and Bartlein (2004) and this 2014 post by Ed Hawkins et al. https://www.climate-lab-book.ac.uk/2014/end-of-the-rainbow/). I can only make this a suggestion for improvement but it's ultimately up to the authors/journal as many papers/journals are still using this colour scheme.*

We appreciate the reviewer´s suggestion.

*19. Further refinement of figure axes and more descriptive captions would be beneficial: For example, what are the units (if any) in Figure 2; adding "t-5, t-4, … t" to the y-axis in Figure 3 would be more visually impactful.*

There are no units in Figure 2. We will try to address this issue in the revised version.

*20. Figure 15 could be combined within Figure 1.*

We believe that it would be hard to convey the message if the figures are combined.

*Mostly well written but would benefit from a final proof read to tidy-up grammar – (i.e. abstract and summary and conclusions, in particular, tense in Section 3 should be in past tense).*

We will address it in the revised version.

We thank the reviewer for the other minor comments that we will fully address in the revised version of the manuscript.

---

## Author Comment (AC3) · 13 Jun 2017

We would like to thank the reviewer for the constructive and thoughtful comments. We will address all the comments in the revised version. Our responses to the comments raised by the reviewer (_underlined and in italic_) are provided below.

_Please provide additional discussion on how the proposed approach help short-term flood forecasting._

This was discussed along lines 33-35 (page 15) and lines 1-3 (page 16). In the revised version, we will extend the discussion.

_Please provide assumptions and limitations of the proposed approach, mainly due to SOM approach and using ERA-Interim reanalysis datasets._

The main assumption of this work is stated along lines 28-33 of page 2. The proposed approach is limited by how well the ERA-Interim reanalysis datasets represent the actual atmospheric circulation and ocean SST, but, due to the lack of data in the southern hemisphere, it is hard to evaluate this limitation.

_Clarify "typically of some km2" in line 7 of page 3._

It is in the order of $10-10^2$ kilometers.

_Please provide justification or the limitations of using 2.5o x 2.5o resolution ERAInterim reanalysis datasets (moisture fluxes, vorticity, upper level winds and sea surface temperature) in the proposed approach for investigating the basin scale flood mechanisms._

At this stage we are more interested in finding general large-scale patterns of atmospheric circulation and SST anomalies associated with extreme floods in the UPRB. For this reason, we believe that such resolution provides satisfactory results.

---

## Author Response (AR1)

We would like to thank the editor for the constructive comments that greatly improved the manuscript. Our responses to the comments (_underlined and in italic_) are provided below.

**General Comments:**

_In this manuscript, flooding is investigated with the focus on the flood generating mechanisms related to precipitation. As the ESD is a interdisciplinary journal I suggest that the authors add some additional information in the introduction on other possible flood generating mechanisms, but highlighting that are not being further investigated in this study._

We agree; We have briefly mentioned it on p. 3, lines 17-18.

_In the entire manuscript, please adjust how the references in parentheses are given (i.e. instead of (Reference et al. (XXXX)) or (Reference (XXXX)) use (Reference et al.,XXXX) and (Reference, XXXX) respectively. Please also see http://www.hydrology-and-earth-system-sciences.net/for_authors/manuscript_preparation.html 'References' for more details on referencing formats and for the 'Examples for reference sorting'._

We have adjusted all the references in the manuscript.

_Throughout the manuscript, the authors introduce and describe new methods. To make the workflow of the analysis clear to the reader, please compile all methods used in the study in one single methods section. In the new methods section, I recommend that the authors describe their workflow in detail and how the individual analysis steps link together, as currently the train of thought is not clear._

All the methods and the workflow of the paper are now presented in the revised Technical Approach section. The revised section provides a detailed description of the methodological framework.

_Additionally, particular attention should be given to the physical reasoning behind the application of a specific method to scientific problem that is being addressed. The authors should communicate to the reader why the chosen method is applied in the current setting and what are the advantages, caveats, and limitations of the method in the current setting._

We thank the Editor for the suggestion. We have tried to address this issue as best as we can, given the state of knowledge. We note that there are some empirical findings that need to be understood in the context of the processes, which will require more similar analyses to be performed in the future.

*I think the manuscript would benefit if the results obtained would be discussed with a stronger focus on the physical interpretation and meaning. This could be done, for example, in a separate discussion section in which the results would also be discussed in the setting of the broader literature.*

We thank the editor for this suggestion. We have tried to discuss all the results obtained in the manuscript with a physical interpretation, including the appropriated citations. We believe the results and summary/conclusions sections are suitable for a broader audience, including hydrologists without a formal background in atmospheric and ocean physics.

*Comments to Figures:*

*The manuscript contains several very interesting and informative Figures. To maximise their readability and information transfer, I suggest the following changes to the Figures:*

*- Please use in all Figures that use spatial data the same geographical projections to allow for easier comparison and evaluation of the different Figures.*

We agree. We have kept the same projections when they have similar purpose (e.g. moisture transport and high level circulation). Please note that the figures are produced by different software according to their purpose.

*- Please make sure that all figures are also legible for someone who is colour blind (i.e. avoid the use of red and green colour in the same Figure). On this see also (http://www.earth-system-dynamics.net/for_authors/manuscript_preparation.html Point 7 under 'Figure composition'. For example on the website http://colorbrewer2.org you can pick colour palettes that are 'colour-blind safe', or you might want to check your Figures using various online tools (for example: http://vischeck.com/vischeck/vischeckImage.php). On this please also the http://www.earth-system-dynamics.net/for_authors/manuscript_preparation.html*

We have changed figures 4 to 8 to comply with this request.

*- Additionally, when using the same colours in several different Figures (particularly if they are meant to be interpreted together), please make sure that the colour at the zero level is the same to allow for a meaningful comparison (e.g. by using diverging colour schemes).*

We agree and we have addressed this issue in the revised version.

*-Finally, please check that all text used in the Figure remains legible when the Figures are being resized for the final publication (i.e. ~8cm for a single column Figure or ~18 cm for double column). E.g. in Figure 2 the size of the text and the numbers need to be increased together*

*with thicker red line or in Fig 14 the line width and the size of the arrow head needs to be increased.*

We agree and we have addressed this issue in the revised version.

*Figure 2: the colour scheme used in the right panel is not intuitive, which makes it difficult to determine if neurons 1 and 3 or neurons 2 and 4 have 'smaller distances'. I suggest using a sequential colour scale instead.*

We have changed this figure and used a sequential color scale as suggested.

*Additionally, I suggest adding the neuron numbers into the blue hexagons in the right panel for ease of interpretation.*

Ok. Done.

*Figure 3: Please add labels to the rows (i.e. t-5, t-4 … t).*

Ok. Done.

*Figure 9: Please add a more detailed legend to Figure that shows either the point size of several percentages or at least the maximum %, 50% and the minimum %.*

Ok. Done.

*Figure 13 and 14: You might want to add the river network in the background of the Figure to allow for easier comparison.*

Unfortunately, these figures are made with a specific package in R (igraph) and the coordinates do not exactly match those of the river network. Besides, the river network in the background makes the visualization of the arrows very difficult.

*Figure 15: I support the suggestion from Referee #2 to combine Figure 1 and 15. Maybe this can be done by adding the sub-basin outlines (without fill colour) and the numbers to Figure 1 as they appear in Figure 15.*

We have combined Figures 1 and 15.

*Specific Comments:*

*In the introduction section, the authors use several times the term 'exceptional floods'. Please add a definition of what this means in the context of the current manuscript.*

Added on line 19 of page 2.

*P3L31: Please specify which of the two basins mentioned above the 'It' refers to.*

Done in the revised version.

*P4L14: For the interdisciplinary readership of ESD, please provide more details on what the 'naturalised' streamflow data entails.*

Addressed on lines 13-15, page 4.

*P4L23: For the interdisciplinary readership of ESD, please provide more details on the threshold selection of the empirical flood quantiles.*

It is not clear for us what additional details we should provide on *"threshold selection of the empirical flood quantiles"*. We will be very happy to provide them if the Editor can be more specific on what is missing in our explanation. As noted, any empirical threshold can be selected for flood analysis.

*Additionally, please elaborate how the 70th percentile is in line with the interest on 'exceptional floods' stated in the introduction section?*

We have addressed this issue in the revised version on page 4, line 18.

*P4L31-33: the procedure described here is not clear to me. Can you elaborate what 'with respect to the day being evaluated' means with regard to the 'long term monthly mean'.*

We removed the term "*with respect to the day being evaluated*" and added the "*respective long term monthly mean*".

*P5L11: On page 4 it is stated that 'naturalized mean daily streamflow data is used. Here the authors now mention river stage as a means of defining a flood event. This might be confusing, consider rephrasing.*

We are just defining the kind of floods we will work on here, in order to avoid confusion with urban floods and storm surges. We have made this clear in the data section indicating we are using discharge data and not river stages.

*P6L3: What are 'tuned parameters'?*

We have addressed this issue. Please see Lines 10-11, page 6.

*P8L30-31: It is not clear to me, why 'neuron 4 likely reflects conditions close to average'? Please elaborate further on this.*

Thanks for bringing this point. We have rephrased this sentence.

*P9L10: 'we do not expect substantial changes…' Is this based on informed hypotheses or was this tested?*

We have removed this sentence and rephrased the paragraph.

*P11L1-2: the 11% given here are based on an original transition probability of 0.117 and the 35% are based on 0.347 as given in the Table 1. It seems that in both cases different rounding rules were used. Please check.*

We have used the same rounding rules in the revised version.

*P11L8: Please elaborate why you have chosen to begin the transition cycle at neuron 3. What is the physical reasoning behind this? Additionally, please highlight at this point in the paper again, that the neurons do most likely not transition.*

There is no specific reason. We can arbitrarily begin the cycle at any neuron. We have changed the sentence and highlighted that the neuron has smaller transition probabilities (lines 23-25, p. 12).

*P 11L16-24: For the interdisciplinary readership of ESD, please elaborate this aspect in more detail with physical explanations in mind.*

We thank the Editor for the suggestion. We have tried to address this issue as best as we can, given the state of knowledge. We note that there are some empirical findings that need to be understood in the context of the processes, which will require more similar analyses to be performed in the future.

*P11L25-P12L18: For the interdisciplinary readership of ESD, please provide a more physical interpretation of the scaling results.*

We thank the Editor for the suggestion. We have tried to address this issue as best as we can, given the state of knowledge. We note that there are some empirical findings that need to be understood in the context of the processes, which will require more similar analyses to be performed in the future.

*P12L6: Please specify what statistical test was used.*

We have specified the statistical test used.

*P12L7-L9: It is not clear to me, how from the analysis conducted one can make interpretations on the 'rainfall intensity'. Please elaborate on this.*

The intensity of rainfall associated with each neuron is showed in Figure 3. We have rephrased the sentence.

*P13L24-P14L1: This sentence is not clear. Please consider rephrasing.*

We have rephrased the sentence.

*P13L21 & 14L2: Please consider rephrasing. The term 'flood propagation' suggest some sort of physical connection and downstream of the flood (i.e. being on the same river channel), which is not the case for all arrows. The same applies to the term 'flow direction' used in this paragraph and the Figure captions.*

In fact, the flood propagates across the basin through a physical connection: either along the same river channel or due to the storm track movement. Hence, we believe that the terms propagations and direction are appropriate in this context.

*P14L5-L15: Please interpret the results more physically, linking the process interpretation to the results from the other sections. In your interpretation you might also want to discuss the time that a typical flood wave generated in the headwaters would need to travel to reach the basin outlet.*

In discussion of the results, we have linked, as best as we can, our findings with the neuron features and provided some physical explanation. We hope that we have addressed the Editor's comment. If not, it would be helpful if the Editor indicates the specific points she wants us to clarify. This is a very large basin and, as we discussed in the manuscript, there are

different patterns of flood routing and we are not sure whether showing a single residence time, which we still do not have available, would help the discussion.

*P14L7: Please specify, that you mean by 'the largest synchronisation'. How is this quantified?*

It is quantified by the number of arrows.

*P 15L20: Please specify, that is meant by 'spatial scaling exponents' in this setting.*

This is the term used in Hydrology to refer to the slope coefficient obtained from the log-log scaling of flood moments (mean, variance, etc) with drainage area. Please see the revised version.

*For the interdisciplinary readership, please make sure that you define all variables e.g. Q on page 13 or Q and A in Figure 11 and 12.*

Done.

*Non-public comments to the Author:*

*Dear Authors,*

*The work that you present in this manuscript is very interesting and well executed. However, the physical reasoning behind the method choices and the interpretation of the results could be strengthened to bring the paper to its full potential. Therefore, I have ask for major revisions to give you enough time to revise the manuscript and to be able to send the manuscript out for review again.*

Again, we would like to thank the Editor and the reviewers for the constructive comments that greatly improved the manuscript. We have tried to address the major points the Editor raised. We hope we have satisfactorily addressed all issues raised.

[revised manuscript text omitted]

---

## Author Response (AR2)

We would like to thank Editor Julia Hall and Reviewer Shaun Harrigan for the additional comments that improved the manuscript. Our responses to the comments raised (*underlined and in italic*) are provided below.

**Comments from the Editor**

*Additionally, please check again that you have addressed ALL comments from the first round of reviews. There are several points that were 'promised to be corrected/amended' in your response to the referees but were never incorporated into the final document.*

We have reviewed the entire manuscript and we hope now that all comments (including from the first round of reviews) are satisfactorily addressed.

*Generally, the word 'somewhat' is used very often in the manuscript. You might want to consider rephrasing for some instances.*

We have removed some instances and rephrased some sentences to address this comment.

*P2 L14: Please specify which processes you are refereeing to.*

It is now specified in the revised manuscript (P2 L14).

*P4 L17: I'm not sure if this reference is appropriate, as the reference given does not contain any information that would assist the reader with further information. I suggest giving the percentage of floods in this season instead. This has already been mentioned by one of the reviewers and it was 'promised' that this would be added but has not been added so far.*

We have removed the reference and provided the percentage of floods during this period (P4 L16 ).

*P5 L11-13. Please rephrase the sentence for clarity and check grammar.*

We have rephrased this sentence.

*P5 L17-22 Please add references to all general statements presented in this section.*

Please note that we are not aware of a specific paper that has addressed this issue. This section is based on our observations from data and our experience in the region. For this reason, we cannot add additional references here.

*P8 L1: Please quantify what 'main rainy season' entails and add reference for that statement.*

We have clarified this in the revised manuscript.

*P8 L6: 'The understanding … will be qualitatively explored'. Please rephrase sentence, currently it reads as if the understanding is explored, which is confusing.*

We agree and we have rephrased this sentence.

*P9 L7-9: Please rephrase sentences. Currently the meaning of the sentence is not clear and/or the grammar seems not correct to me.*

We have rephrased this sentence.

*P9 L30: 'neurons 3 and 4' need to be replaced with 'neurons 3 and 2'*

We have corrected it. Thank you!

*P 10L3-L13: I think it would be beneficial to the reader if the description of the anomalies would focus more on the positive rainfall anomalies. E.g. for t-5 and t-4 for neuron 2 the rainfall field is not just 'somewhat homogenous' but appears to be a larger scale positive anomaly and the progression of negative rainfall anomalies (rainfall less than the long-term average) at later time steps seems to be of less importance for the flood generation. This applies similarly to the other neurons as well.*

We generally agree that the positive anomalies are more relevant to this paper. But, in our opinion, the negative rainfall anomalies constitute a spatial pattern that is worth mentioning in the text. One of the fascinating issues in our study area is (often rapid) changes between dry and wet cycles.  For this reason, we have included few sentences on the negative anomalies.

*P10 L14-L17: This paragraph is somehow repetitive of the first paragraph on this page. I suggest combining the two. Additionally, when looking at Figure 2 I think the last sentence should read 'The shortest distance is obtained between neurons 3 and 4, followed by the distances of neurons 2 and 4 and neurons 1 and 3', to preserve the decreasing order of the distances.*

We have removed this last sentence and rephrased the last sentence of the first paragraph.

*P10 L28 – P11 L14 Please elaborate for the interdisciplinary readership of ESD that a negative divergence field (i.e. convergence) indicates that rainfall can occur (brief description of the processes involved), but please also make the reader aware that rainfall does not necessarily have to occur.*

We have added a statement to the revised paper (P11 L2-3) to address this issue.

*P11 L1: 'negative events' of what? Please elaborate.*

We have removed the term negative events.

*P11 L2: 'The SALLJ is weak' it is not clear to me to which part of the text before this statement refers to. Please clarify.*

We have rephrased this sentence.

*P11 L9 For clarity, please add 'positive' before 'divergence'.*

Ok.

*P11 L11-13: 'Neuron 4 has a moisture transport pattern somewhat similar to that of neuron 2, but the origin of the fluxes are more associated with the South Atlantic, with meridional fluxes west of the basin, and a less intense but still relatively homogeneous moisture convergence.' From Fig. 4, I am not able to come to the conclusions that are written after the word 'but' in the above sentence. The two panels look very similar to me. Please check again and rephrase.*

We have removed the sentence after 'but'.

*P11 L 15-27: Please add a sentence explaining to the interdisciplinary readership the reasons behind choosing to analyse the 850 and the 500 mb (i.e. what different type of conclusions can be drawn at the different levels).*

These two levels (*850 and 500 mb*) are common thresholds for studying large-scale patterns in the region. We have highlighted the different conclusions from these thresholds in the main manuscript (P11, L15-21).

*P 11 L16 Please specify to the reader what kind of wave activity and what dynamic forcing you are refereeing to by better describing the mechanisms involved.*

We refer to divergence in the upper levels. We have clarified this issue in the revised version (P11, L18).

*P 11 L23: The attribute 'through' only applies when specifically talking about a wave, however here it is used in connection with the word 'circulation'. Please rephrase statement for added clarity (see also sentence describing neuron 3).*

We think that the word "trough" can also be used in connection with atmospheric circulation when the flow has a wave kind of pattern. For instance, we observe in many bulletins (e.g. [http://www.cpc.ncep.noaa.gov/products/analysis_monitoring/bulletin_1201/extra.shtml](http://www.cpc.ncep.noaa.gov/products/analysis_monitoring/bulletin_1201/extra.shtml)) and textbooks (e.g. Wallace and Hobbs, Atmospheric Science) the term "trough" used in connection with atmospheric circulation.

*P 11 L24: Please be more specific where 'southwestern of it' is located (similar to how the location was described in detail for Figure 5).*

Ok.

*P 11 L28 replace 'is shown' with 'are shown'*

Ok.

*P 12 L1: For clarity please specify the altitude, where the described 'El Niño region' is located and what definition of Nino is being used. This has already been mentioned by one of the reviewers before but has not been incorporated yet!*

There are different definitions for the El Nino region. For instance, there are several indices (NINO1+2, NINO3, NINO3.4 and NINO4) to indicate regions of the eastern Tropical Pacific that experience positive anomalies in the SST and that we could call an El Nino event. There is also the Modoki El Nino that is based on information from the Central Pacific. In this discussion, we are not referring to a specific definition. Hence, we have added "eastern tropical Pacific" to generally describe the 'El Nino region'.

*P 12 L5 please add 'whether' between 'or' and 'they.*

Done.

*P 12 L8-10: What does '(3)' refer to? Figure 3 ? Additionally please elaborate what 'the results of Doyle and Barros (2002)' were. And please also add the latitude location of the tropics and subtropics for ease of interpretation.*

We have added "Fig. 3" and rephrased the sentence. We have also added the latitude location of tropics and subtropics.

*P 12L19: when rounding the number in Table 1 it is probably '17%' and not '18%'.*

Ok. Thanks.

*P12 L 22: 'neuron 4 (Fig. 3)' should probably read 'neuron 3'?*

We have corrected this sentence.

*P 12L 24: The sequence should be 3 -> 1 -> 2 -> 4 -> 3. His has already been mentioned by one of the reviewers but has not been changed yet!*

Ok.

*P13 L 7 I'm not sure if 'remarkably' is the correct term to use here. From all the analysis done before in the paper this does not seem to be 'remarkable. Please rephrase.*

We have dropped the word "remarkably".

*P 13L19: Suggest replacing 'average flow' with 'average flood flow'*

Ok.

*P 14 L 6: Replace 'the edges the …' with the 'the branches represent the….'*

We have used the exact term employed in the literature of complex networks where the original theory comes from, so we prefer to keep the term "edge". But to clarify, in parenthesis, we have added the term branches.

*P 14 L 10-20 You might want to consider linking this also to Fig 3, by discussing the location of the areas with strong positive rainfall anomalies.*

We thank the Editor for the suggestion and this has been linked by mentioning the neurons.

*P15 L6 replace 'peak' with 'peaks'?*

Ok.

*P16 L 14: Please make sure that the link provided is working and that the link directs the reader directly to data source. Currently the link seems broken…*

We have double checked and the link is directing to the data source at the time of this revision.

*Figure 1 please change colour scheme, as the red – green combination of colours will make it difficult to understand the Figure for people that are colour blind.*

All of our maps have been tested for back and white print. Please note that the colors are not green and red. They are green and pink and they are distinctly different in back and white print. Also, the choice of color is not a significant part of the story. If someone is color blind then they will have potential problems with all color figures.

*Figure 2: Please increase either the size of the red coloured numbers (both Figures same font size) or chose a different colour that has a better contrast with the grey surrounding, as currently the red numbers will be difficult to decipher once the Figure is scaled the correct size used for publication.*

We have updated the figure as suggested.

*Figure 4-8: Please make sure that the 'zero' on Figs. 4-8 is the same colour (e.g. white), so the Figures are easily interpretable (i.e. it does not make sense that the zero line goes through the blue shading when a divergent colour scale is being used (e.g. in Fig. 4)). Additionally, to avoid confusion, please make sure that the colour code (i.e. red is negative & blue is positive) is the same for all of the above Figures (currently Fig 7 & 8 are the other way around).*

We thank the Editor for these suggestions. We have updated the figures as suggested.

*Figure 4: Additional to the changes mentioned above, please provide a proper legend (with at least 3 arrows of different length) for the scale of the moisture fluxes, as currently one does not know how the arrows are scaled. Please also add to the Figure caption, that the arrows show the moisture fluxes and the colours the divergent field.*

Please note that we are more interested in the directions of the fluxes rather than its magnitude, which is more important for displaying divergence fields. For this reason, the scale of the arrows is not so relevant for the interpretation of the figure. We believe that one arrow (as is commonly done in the literature) showing the scale (right bottom side of neuron 4) is sufficient to provide an idea of the magnitude of the moisture fluxes. This also avoids overcomplicating the figure.

*Figures 5,6,8 Please add the zero line.*

We have included zero line in Figure 8. We don´t think that adding a zero line to Figures 5 and 6 will provide more information.

*Figure 13 & 14: The arrows are not visible, particularly in Fig 14. Please make them clearly visible (maybe a different colour for the arrow head or increasing the size of the arrow head might help).*

*Figure 14: You might want consider adding a box around the networks for clarity.*

*Additionally, once the Figure is scaled the correct size in the final publication, I suspect that the font used to identify the neurons might be too small. Please increase font size of the labels.*

We agree and we have increased the size of the arrow heads and labels and added boxes to the panels.

*General Comment on References:*

*Please make sure that the references contain all necessary information and are written as outlined in: https://www.earth-system-dynamics.net/Copernicus_Publications_Reference_Types.pdf*

*Please make sure that all journal names are abbreviated according to the ISI Journal Title Abbreviations Index (https://www.library.caltech.edu/journal-title-abbreviations). For example, 'Journal of Hydrology' should be written 'J. Hydrol.'*

We have corrected all references to comply with these requirements.

**Reviewer # 2:**

*The revised manuscript is improved over the original submission and I recommend it for publication in ESD. It is an interesting paper and makes an important contribution to the literature on flood hydroclimatology. I have a few minor suggestions that the authors may wish to take on board:*

*Pg3;L9-10 &Pg4;L13-14: range of drainage areas repeated, remove one.*

We have removed one.

*Pg 4; 17: As per my original comment, can't see the % number of total annual floods that occur Nov-March here?*

We have added the % number of total floods that occur Nov-Mar.

*Pg4; 21-22: As per my original comment: "events with inter-arrival times larger than 15 days, which we believe is a consistent interval to guarantee independence between flood events considering the different rainfall mechanisms that cause floods in UPRB". Unless a formal independence criteria was applied, which there is no mention?, then there is no guarantee.*

Please note that here we are not investigating/sampling individual flooding events here. Our focus is on independent **atmospheric** mechanisms responsible for flooding. It is very unlike that the same rainfall mechanism will persist for more than 15 days and cause multiple major floods. In fact, most mechanisms responsible for the rainfall in the region last less than 3 days (P7, L24 and references therein). Hence, we believe that inter-interval times beyond 15 days are sufficient to guarantee independence.

*Interesting plot in response to my query on the range of peaks-over-threshold events. It might be of interest to other readers to include the range from 76-131 within Pg4;L23 of the revised manuscript.*

We have added the range to the revised manuscript.

[revised manuscript text omitted]

---

## Author Response (AR3)

We thank again Editor Julia Hall for the additional comments. Our responses are provided below.

**Comments from the Editor**

*The authors have satisfactory addressed the comments from the last round of reviews. There are only a few corrections/comments before the paper can be accepted for publication.*

*General Comment:*

*Please make sure that either Parana or Paraná is used consistently in the text.*
We have used "Paraná" along the entire manuscript.

*Detailed comments*

*P2L29: Please check the sentence, due to the correction in the last revision the sentence structure has become corrupted.*

Corrected. Thanks.

*P4L16: 'about 98 flood events (ranging from 76 to 131 events)' This is not clear, is 98 the average for over all sites and the range is given per site? Please specify.*

It is the average across all sites, ranging from 76 to 131. We have rephased the sentence in the revised manuscript.

*P4L 26: Add the values of 'mb' used for the low and high level vorticity.*

We didn´t identify the need to add mb in P4L26. Perhaps the Editor meant to refer to P4L32. But even in this case, we think that the term "vorticity" does not require the use of mb.

*P4L27: 'It covers …. and are…' Please correct sentence structure/grammar.*

Ok. Thanks.

*P5L 9 to 11: Please correct sentence structure/grammar.*

Corrected. Thanks.

*P8L 7: Please specify what 'm' is.*

Done.

*P9L 25 to 28: The figure has changes since the last version, so now 'left panel' or 'right panel' should be replaced by 'top' or 'bottom' panel.*

Done.

*P9 L 29-30. Suggest swapping the order of 'neurons 1 and 3' and 'neurons 2 and 4' to maintain the increasing order of distances.*

Done.

*P10L 11: 'some connections' Please specify/ elaborate.*

Done.

*P10L31: suggest adding 'slightly' before 'positive' or something similar to indicate that the southern part of the basin is not very positive.*

Done.

*P10L33: 'also observed in …' Yes, but in Fig 4 the dipole exists in the larger scale but not so much in Fig3.*

Yes, it appears as large scale in Figure 4 and also covers a large portion of the basin (Figure 3, Neuron 1 at time t).

*P11L8-10: Please make sure that the sentence conveys that the values in the south have almost no positive values in the basin.*

We thank the Editor for the suggestion but we think that the positive values across the basin are in the same range as in other parts of the country, so we prefer to keep as it is.

*P11L10: 'The moisture divergence pattern is again similar to the rainfall field at time t for neuron 3…' I 'm not able to see this. Please be more precise in describing what you mean.*

We have added a sentence in the revised manuscript.

*P11L20: 'also shows cyclonic rotation…' the values in the Figure are very small, please indicate in text.*

We thank the Editor for the suggestion and we agree that the values are small, but we think that it is not necessary to indicate it in the text.

*P11 L22-23: The trough is not only 'weaker' but appears to be very weak to almost non existing in the south of the basin. I suggest adjusting the language to convey this fact. (Maybe adding a zero line to the map in Fig 6 would help in interpreting this).*

We have removed the word "strong".

*P11 L24: 'centered around 45…' Does this refer to the positive or the negative values. Please specify..*

Positive vorticity. The negative vorticity occurs over the entire basin, as indicated in the text.

*P11L26: 'negative vorticity appears only in the south' The negative values are very low and almost nonexistent. Please adjust text. And replace 'kind of' with a more specific statement (this also applies to P12L1)*

We have used the term "slightly", replaced "kind" by "type" a removed "kind" on P12L1.

*P11L30: 'also a sharp contrast'. Please adjust text, as the values are small and do not contrast shapely.*

We have removed the term "sharp".

*P 11: Figure 7 is only covered by about 4 lines of text. I suggest elaborating a little more on the causes of the strong anomalies and sharp contrasts mentioned instead of just describing the Figure.*

We understand the concern of the Editor but we believe that further analysis, including the causes of these anomalies, is beyond the scope of this paper and should be addressed in future work.

*P12L 9: suggest either using 'south of 40…' or 'around 50…'.*

Done.

*P12L13: 'Combining all the analyses…' Please specify which analyses are combined to reach the conclusions of the transition probabilities, as it seems that not all analyses are used.*

We have replaced "all the" by "the preceding".

*P 24: Suggest making the red catchment boundaries in the inset map of Fig 1 thicker as they will be hardly visible when the Figure is adjusted to the smaller publication size.*

We thank the Editor for the suggestion.

*P 25: Add to the Figure caption that the red numbers correspond to the number of the neuron.*
Done.

*Figures: Please correct the colour scale for Figure 3-4 and 6-7 to reflect a symmetric colour scale with similar linear increase for both negative and positive values.*

We have corrected Figures 3, 4, 6 and 7 to address this issue.

*References: Please remove/correct the several references containing 'n/a-n/a'*

Done.